# Bridging the Perceptual Gap: Residual-Enhanced Downscaling and Manifold-Aware Perception Alignment Adaptation for NR-IQA

**Yu Li** [1]  **Zhengran Shen** [1]  **Yachun Mi** [1]  **Puchao Zhou** [1]  **Shaohui Liu** [1]

## Abstract

Leveraging Large Vision-Language Models like CLIP has recently set new benchmarks for No-Reference Image Quality Assessment (NR-IQA). However, the contrastive pretraining of CLIP inherently prioritizes semantic invariance, which often suppresses subtle perceptual signals, a phenomenon we term perceptual submergence. Furthermore, standard preprocessing techniques (e.g., cropping and interpolation) further exacerbate the loss of critical high-frequency quality cues. In this paper, we propose the Cross-modal Perception Alignment Adapter (CMPA), a manifold-aware framework designed to disentangle perceptual distortions from dominant semantics. CMPA introduces a Perception-Sensitive Feature Extractor (PFE) that projects CLIP features into a compact, low-dimensional subspace, explicitly magnifying distortion-induced off-manifold deviations. Subsequently, a Cross-Modal Perception Alignment Injector (PAI) aligns these features with quality-aware text anchors and re-injects them into the backbone. To ensure input fidelity, we also devise a Residual-enhanced Perceptual Downscaling strategy that adaptively compensates for resolution-induced information loss using Just Noticeable Difference (JND) guided frequency re-injection. Extensive evaluations on several benchmark datasets demonstrate that our approach significantly outperforms state-of-the-art methods, effectively recovering the perceptual signals submerged in semantic-dense representations.

## 1. Introduction

No-reference image quality assessment (NR-IQA) is a fundamental yet challenging task in computer vision, aiming

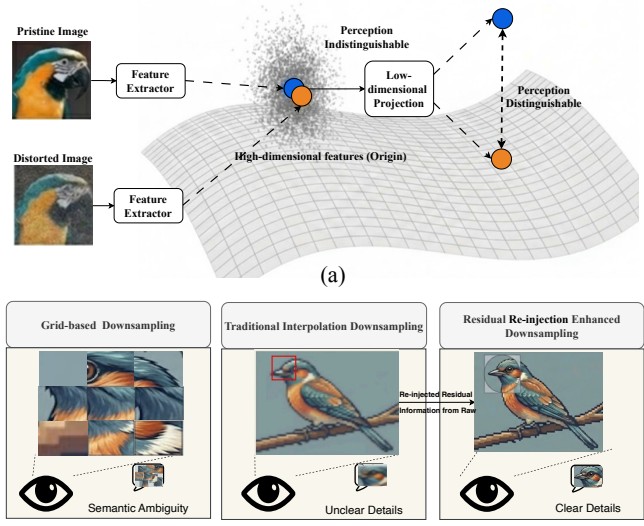

*Figure 1.* Motivation of the proposed method. (a) Low-dimensional projection uncovers perceptual discrepancies by mapping pristine features onto a manifold while isolating distorted samples as outliers. (b) Our residual re-injection compensates for perceptual information loss inherent in conventional downsampling, achieving thumbnails with both semantic clarity and structural fidelity.

to perceive visual degradations without access to pristine references. The task is particularly arduous for real-world images from User-Generated Content (UGC), which are afflicted by diverse and uncontrolled distortions such as motion blur, sensor noise, and mixed compression artifacts. While early works relied on handcrafted statistics (Mittal et al., 2012a;b; Saad et al., 2012) and CNNs (Su et al., 2020), recent state-of-the-art methods have pivoted towards leveraging Vision-Language Models, specifically CLIP (Radford et al., 2021), to harness rich semantic priors for quality prediction (Wang et al., 2023; Zhang et al., 2023).

Despite their success, we argue that a critical theoretical disconnect persists in existing CLIP-based methods—ranging from early explorers (Wang et al., 2023; Zhang et al., 2023) to recent zero-shot frameworks like QualiCLIP (Agnolucci et al., 2024), typically treat CLIP embeddings as monolithic feature pools. These approaches typically treat pretrained CLIP embeddings as generic feature pools, overlooking

[1]Harbin Institute of Technology, Harbin, China. Correspondence to: Shaohui Liu <shliu@hit.edu.cn>.

*Proceedings of the 43rd International Conference on Machine Learning*, Seoul, South Korea. PMLR 306, 2026. Copyright 2026 by the author(s).

the fundamental *distributional characteristics* of the feature space. CLIP is optimized for semantic alignment via contrastive learning, which explicitly encourages *semantic invariance*—the model is trained to map images with the same content (e.g., a sharp bird and a blurry bird) to the same semantic concept. Consequently, in the original high-dimensional hypersphere, perceptual distortion signals are effectively treated as "non-semantic noise" and suppressed.

As a conceptual illustration in Fig. 1(a), we hypothesize a phenomenon termed 'perceptual submergence', where subtle distortion-induced variations are masked by dominant semantic features in the high-dimensional embedding space. In the original high-dimensional space, features of pristine (blue) and distorted (orange) images cluster tightly based on content, rendering quality-related deviations mathematically indistinguishable from semantic variations. However, our preliminary observation in Fig. 1(a) also suggests that when these features are projected into a compact subspace, the distortion-induced off-manifold perturbations become significantly more distinguishable.

Inspired by the Manifold Hypothesis (Bengio et al., 2013), we propose the Cross-modal Perception Alignment Adapter (CMPA). Our key insight is that while CLIP features are dominated by high-dimensional semantic manifolds, perceptually salient cues reside in a more compact, separable low-dimensional subspace. CMPA disentangles these cues through two synergistic components: (1) the Perception-Sensitive Feature Extractor (PFE), which projects high-dimensional embeddings into a bottleneck subspace to filter out semantic redundancies and expose latent perceptual manifolds; and (2) the Cross-Modal Perception Alignment Injector (PAI), which aligns these visual residuals with quality-aware text anchors. This low-dimensional design is empirically corroborated by findings in LoDA (Xu et al., 2024), where IQA performance was found to peak at a remarkably low intrinsic dimension (e.g., d=64), further validating the efficacy of our projection strategy. MA-CLIP (Liao et al., 2025) also identified that CLIP's semantic-dense features tend to overlook critical quality cues, further justifying our motivation to disentangle perceptual signals from the dominant semantic manifold.

Our framework also addresses the critical loss of perceptual cues often overlooked during mandatory input resizing. To fit fixed input constraints, practitioners often resort to grid-based cropping or traditional interpolation. As analyzed in Fig. 1(b), grid-based methods suffer from *Semantic Fragmentation* (Wu et al., 2022), while traditional interpolation induces *Perceptual Aliasing* (Parmar et al., 2022), indiscriminately smoothing out high-frequency textures that are vital for quality discrimination. To resolve this, we introduce a **Residual-enhanced Perceptual Downscaling (RPD)** preprocessor. By decomposing the raw image into

base and residual frequencies, we adaptively re-inject lost high-frequency details into the downsampled input, guided by Just Noticeable Difference (JND) weighted map. As conceptually illustrated in Fig. 1(b), RPD is designed to preserve the clear details of the original capture, aiming to prevent the model from being misled by artifacts typically introduced by traditional resizing. In summary, our contributions are threefold:

1. We propose CMPA, a manifold-aware adapter comprising a Perception-Sensitive Feature Extractor and a Cross-Modal Perception Alignment Injector. This architecture effectively disentangles subtle quality cues from dominant semantic noise by projecting features into a low-dimensional perceptual subspace and subsequently re-injecting aligned perceptual anchors into the backbone.

2. We introduce a RPD preprocessor that adaptively compensates for resolution-induced information loss using frequency separation and JND-guided residual injection. This approach resolves the perception-distortion mismatch caused by traditional resizing, ensuring the model's predictions are grounded in the intrinsic quality of the raw image rather than preprocessing artifacts.

3. Extensive evaluations on several benchmark datasets demonstrate that our framework consistently outperforms SOTA methods, validating the effectiveness of our methods.

## 2. Related Works

### 2.1. Learning based Image Quality Assessment

Deep learning has revolutionized NR-IQA by directly mapping image content to perceptual quality scores. Early approaches based on Convolutional Neural Networks (CNNs) typically employed pre-trained networks, such as ResNet (He et al., 2016), as backbones to leverage hierarchical feature extraction for quality prediction. Hyper-IQA (Su et al., 2020) introduced adaptive hyper-networks to decouple the IQA process into content understanding and quality prediction stages, thereby mimicking the content-sensitive characteristics of the Human Visual System (HVS). To capture long-range dependencies, Transformer-based methods (Vaswani et al., 2017), exemplified by MUSIQ (Ke et al., 2021) and MANIQA (Yang et al., 2022), utilize multiscale attention mechanisms to facilitate interaction between global and local regions. Concurrently, self-supervised methods (Madhusudana et al., 2022; Zhao et al., 2023) leverage contrastive learning to mitigate the scarcity of labeled data; however, they often struggle to bridge the domain gap between synthetic pre-training tasks and authentic distortions. Recently, Vision-Language Models (VLMs) have shifted the paradigm from pure visual regression to semantic-assisted quality assessment. Wang et al. (Wang et al., 2023) explored the zero-shot capabilities of CLIP but encountered misalignments between high-level semantics

and low-level distortion patterns. Although subsequent fine-tuning strategies (Zhang et al., 2023; Li et al., 2025; Mi et al., 2024) incorporated scene and distortion priors, they typically process visual and textual branches independently prior to fusion. Crucially, these methods lack deep, layer-wise interaction mechanisms, failing to explicitly guide image features to query quality-relevant definitions from the text encoder, thereby limiting fine-grained alignment. In this work, we take advantage of well-pretrained VLMs and adapt them via layer-wise interaction between visual features and quality-aware linguistic priors.

## 2.2. Efficient Model Fine-Tuning Methods

In recent years, visual foundation models have witnessed an exponential surge in parameter scale, evolving from early EfficientNet (0.48B parameters) (Pham et al., 2021) to contemporary Transformer-based massive models reaching 22B parameters (Dehghani et al., 2023). The shift from full fine-tuning to Parameter-Efficient Fine-Tuning (PEFT) has become essential as foundation model scales escalate (Dehghani et al., 2023). Current PEFT methodologies primarily bifurcate into two streams: Approaches like LoRA (Hu et al., 2022) and FacT (Jie & Deng, 2023) inject trainable low-rank matrices to adapt to downstream tasks. However, these methods treat the low-rank subspace primarily as an optimization tool, lacking a dedicated mechanism for perceptual feature analysis. Methods such as Adapt-Former (Chen et al., 2022) and ConvPass (Jie & Deng, 2022) insert lightweight bottleneck or convolutional modules. Recently, (Xu et al., 2024; Li et al., 2026) further validated the feasibility of employing such Transformer adaptation techniques to enhance IQA performance. Nevertheless, being designed for semantic reconstruction, these generic adapters tend to suppress subtle feature perturbations induced by image distortions. In contrast, our work moves beyond generic modulation by introducing a specialized mechanism designed to decouple quality-sensitive cues from dominant semantic features within a learned manifold structure.

## 2.3. Perceptual Image Pre-processing Strategies

Modern IQA backbones typically require fixed-size inputs, necessitating strategies to balance perceptual detail with semantic integrity. Current methodologies primarily rely on patch-based sampling to handle high-resolution inputs. Early works such as MUSIQ and FastIQA (Wu et al., 2022) extract multiple local patches to preserve fine-grained fidelity. To further capture scale-variant distortions, some works (Liu et al., 2024; Mi et al., 2025) introduces a multi-resolution patch sampling strategy, aggregating girds from different scales to enrich the representation. While these methods effectively mitigate the information loss of a single resolution, the discrete nature of patch-based approaches inevitably disrupts the global semantic layout—a critical

prior for Vision-Language Models like CLIP, and often requires complex aggregation mechanisms. Alternative adaptive sampling or downsampling methods aim to maintain the global structure, yet standard bicubic operators or rate-distortion-driven approaches (Liang et al., 2024) often suppress quality-aware high-frequency details like subtle noise or blur. In this work, we propose Residual-enhanced Perceptual Downscaling (RPD) to perform unified downsampling while adaptively re-injecting lost perceptual cues, ensuring a holistic and quality-aware input for assessment.

## 3. Methods

### 3.1. Overview

The overall architecture of our proposed framework is illustrated in Fig. 2. We introduce a parameter-efficient paradigm for BIQA, leveraging a frozen CLIP backbone augmented with our Residual-enhanced Perceptual Downscaling (RPD) and Cross-modal Perception Alignment (CMPA) modules.

To adapt raw high-resolution inputs to the fixed resolution required by pretrained encoders without sacrificing quality-sensitive details, we first apply the RPD preprocessor. Unlike standard interpolation that indiscriminately smooths out high-frequency transients, RPD explicitly captures and re-injects residual textures from the original image into the downsampled representation. This ensures that the backbone receives a perception-consistent input where critical distortion cues are preserved.

The downsampling image, alongside quality-related linguistic prompts such as ["Low quality image", "High quality image"], is then fed into the frozen CLIP image and text encoders. To address the phenomenon of perceptual submergence, we insert CMPA adapters across the transformer blocks to decouple perceptual signals from the dominant semantic manifold. Within each CMPA, the Perception-sensitive Feature Extractor (PFE) projects high-dimensional features into a compact, low-dimensional bottleneck subspace—a design motivated by the hypothesis that the intrinsic dimension of perceptual quality is significantly lower than that of semantics. Subsequently, the Perception Alignment Injector (PAI) facilitates deep cross-modal interaction via bidirectional cross-attention, calibrating the visual features with quality-aware linguistic priors. Finally, the quality score is derived by computing the cosine similarity between the aligned visual global output and the textual quality anchors.

### 3.2. Perception-Sensitive Feature Extractor

To expose the subtle perceptual deviations often diluted in high dimensional semantic spaces, the PFE explicitly maps the multi-modal embeddings into a compact latent bottleneck. Let $F_{img} \in \mathbb{R}^{L \times D}$ and $F_{txt} \in \mathbb{R}^{2 \times D}$ denote

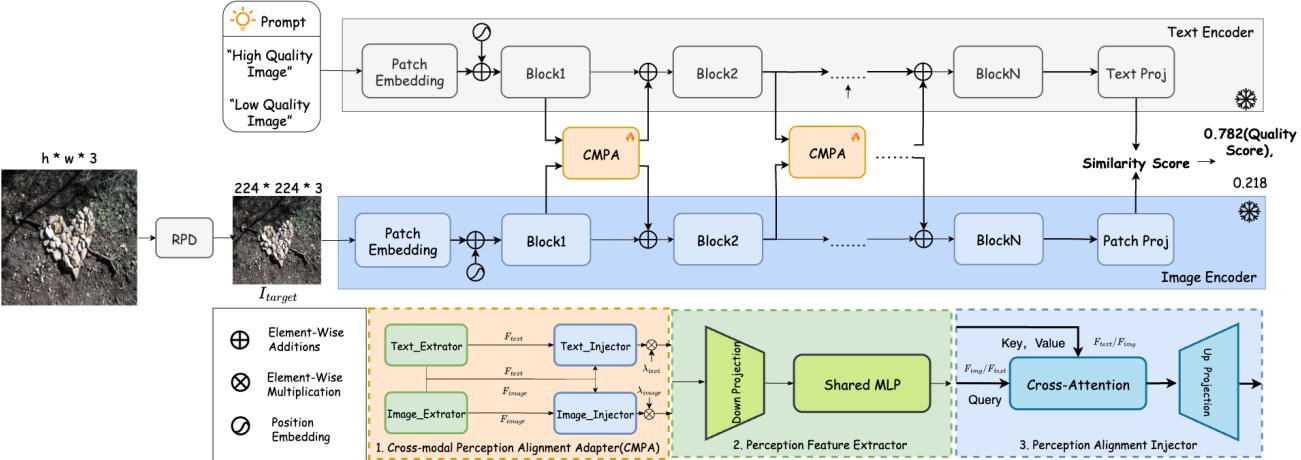

*Figure 2.* Overall architecture of the proposed framework. Given an input image at original resolution, the RPD is first employed to produce a downsampled version that fits the backbone's input constraints. Subsequently, the CMPA is integrated into the frozen CLIP blocks via a layer-wise scheme. Specifically, CMPA extracts and aligns perceptual features in a low-dimensional manifold through cross-modal guidance, which are then re-injected into the original CLIP features. The final image quality assessment is performed by computing the similarity score between the perception-aligned image embeddings and text prompts.

the visual and textual features from the frozen CLIP blocks. We first perform a dimensionality reduction to project both modalities into a low-dimensional subspace $d \ll D$:

$$Z_m = \text{Linear}_{D \to d}(F_m), \quad m \in \{\text{img}, \text{txt}\} \quad (1)$$

This projection serves as a structural bottleneck that filters out redundant semantic invariants. To facilitate initial cross-modal synergy within this subspace, we employ a Shared MLP to refine the projected features:

$$\hat{Z}_m = \sigma\big(W_2\,\sigma(W_1 Z_m)\big) \quad (2)$$

By enforcing weight sharing across modalities, the Shared MLP acts as a modality-agnostic perceptual filter, forcing the bottleneck to capture unified distortion patterns rather than modality-specific noise.

### 3.3. Cross-Modal Perception Alignment Injector

Building upon the distilled bottleneck representations, the PAI stage executes deep cross-modal alignment to ensure that the extracted manifold deviations are perceptually consistent across both modalities. As illustrated in the detailed CMPA architecture, we employ a dual-path multi-head Cross-Attention(MHCA) mechanism to facilitate mutual modulation between visual and textual features.

The interaction in the $d$-dimensional subspace is defined by two complementary paths, where one modality acts as the query to calibrate the other: the visual features $\hat{Z}_{\text{img}}$ serve as the query, while the textual anchors $\hat{Z}_{\text{txt}}$ provide the key-value pairs to guide the perceptual alignment. The refined visual feature $F_{\text{img}}^{\text{align}}$ is formulated as:

$$F_{\text{img}}^{\text{align}} = \text{MHCA}(\hat{Z}_{\text{img}},\ \hat{Z}_{\text{txt}},\ \hat{Z}_{\text{txt}}) \quad (3)$$

Symmetrically, the textual prompt embeddings query the visual patch features to capture local, distortion-sensitive cues, yielding the aligned textual feature $F_{\text{txt}}^{\text{align}}$:

$$F_{\text{txt}}^{\text{align}} = \text{MHCA}(\hat{Z}_{\text{txt}}\ \hat{Z}_{\text{img}},\ \hat{Z}_{\text{img}}) \quad (4)$$

This dual-path mechanism allows the linguistic quality anchors to dynamically "supervise" the visual extraction while simultaneously updating the text embeddings with image-specific distortion evidence.

Following the cross-modal interaction, the aligned features are projected back to the original $D$-dimensional space. To preserve the robust semantic priors of the CLIP backbone while augmenting it with perceptual sensitivity, we apply a residual-style injection:

$$F_m^{\text{final}} = F_m + \text{Linear}_{d \to D}(F_m^{\text{align}}), \quad m \in \{\text{img}, \text{txt}\} \quad (5)$$

where $F_m$ denotes the original high-dimensional features from the frozen encoders. By re-injecting these refined "perceptual residuals," the framework effectively bridges the gap between high-level semantic understanding and low-level degradation sensitivity.

### 3.4. Residual-Enhanced Perceptual Downscaling

To mitigate the fidelity loss inherent in standard resizing, we propose Residual-enhanced Perceptual Downscaling (RPD). As illustrated in Fig. 3, RPD rectifies the information deficit by decomposing the raw signal into multi-frequency components and re-injecting sampling residuals extracted via a Residual Enhanced Downscaling (RED) module. Given an input component $\mathbf{I}_{\text{in}}$, the RED operator $\mathcal{R}(\cdot)$ extracts the sampling residual by measuring the deviation between the

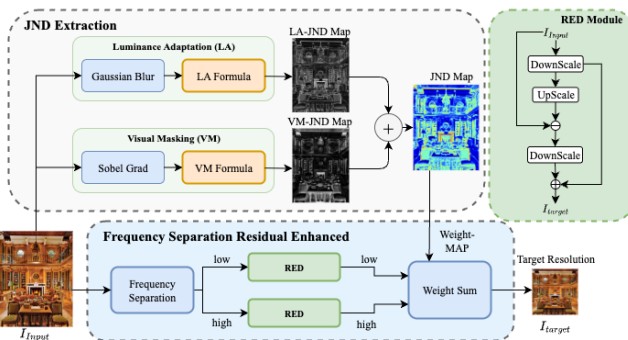

*Figure 3.* Architecture of the RPD. Input images are decomposed into frequency-specific components and processed via the RED module to extract structural residuals. These residuals are then adaptively fused using a JND-guided weight map to generate the perceptually-consistent output $I_{target}$.

original signal and its reconstruction through a down-up sampling cycle:

$$\mathcal{R}(\mathbf{I}_{in}) = \text{down}(\mathbf{I}_{in}) + \text{down}(\mathbf{I}_{in} - \text{up}(\text{down}(\mathbf{I}_{in}))) \quad (6)$$

where $\text{down}(\cdot)$ and $\text{up}(\cdot)$ denote the downsampling and upsampling operators, respectively. These residuals $\mathbf{I}_{in} - \text{up}(\text{down}(\mathbf{I}_{in}))$ explicitly represent the high-frequency details and structural information that are typically diluted in standard downsampling.

To ensure the residual injection is perceptually optimal, we introduce an adaptive gain map $\mathbf{A}$ to modulate the injection intensity. We first estimate a Just Noticeable Difference (JND) map $\mathbf{M}_{jnd}$ based on luminance adaptation $F_{la}$ (Chou & Li, 1995) and visual masking $F_{vm}$ (Yang et al., 2005):

$$\mathbf{M}_{jnd} = \frac{1}{2}\Big(F_{la}(L_m) + \beta \cdot F_{vm}(G)\Big) \quad (7)$$

where $L_m$ denotes the local mean luminance estimated via Gaussian smoothing, and $G$ is the texture gradient magnitude computed using Sobel operators. $\beta$ is a weighting coefficient. To optimize residual integration, we introduce an adaptive gain map A to modulate the injection intensity. Grounded in the Weber-Fechner Law, which characterizes the logarithmic nature of human visual perception, the gain is formulated as:

$$\mathbf{A} = 1 + \log_2\left(1 + \left(1 - \frac{1}{d}\right)\right) \cdot (\mathbf{M}_{jnd} \cdot \alpha) \quad (8)$$

where $d$ is the downsampling ratio and $\alpha$ is a scaling factor. The term $\log_2(2 - 1/d)$ serves as a differentiable weight within [0,1) that monotonically increases with the $d$. This ensures the injection intensity adaptively compensates for information loss, prioritizing residuals in regions where the HVS is most sensitive to distortions.

Finally, RPD performs frequency separation by applying a Gaussian filter $G(k, \sigma)$ to decompose the raw image into low-frequency $\mathbf{I}_L$ and high-frequency $\mathbf{I}_H$ components. The target representation $\mathbf{I}_{target}$ is resynthesized by recursively aggregating the base downsampled features with the residuals extracted by the RED operator $\mathcal{R}(\cdot)$:

$$\mathbf{I}_{target} = \mathcal{R}(\mathbf{I}_L) + \mathbf{A} \odot (\mathcal{R}(\mathbf{I}_H)) \quad (9)$$

where $\odot$ denotes element-wise multiplication. By leveraging the RED-derived residuals, RPD provides a high-fidelity input that bridges the gap between semantic understanding and distortion sensitivity. Further rationale of the feasibility of RPD can be found in Appendix. B.

## 4. Experiments

### 4.1. Experimental Settings

**Datasets**: Our method is evaluated on classical IQA datasets, including four synthetic datasets, LIVE (Sheikh et al., 2006), CSIQ (Larson & Chandler, 2010), TID2013 (Ponomarenko et al., 2015), KADID-10k (Lin et al., 2019) and four authentic datasets, LIVEC (Ghadiyaram & Bovik, 2015), KonIQ-10k (Hosu et al., 2020), SPAQ (Fang et al., 2020), and FLIVE (Ying et al., 2021).

**Implementation details**: To ensure a fair comparison, we follow the experimental setup of LoDa (Xu et al., 2024). A key distinction in our approach is the omission of image resizing to better evaluate the effectiveness of the proposed RPD and alignment-amplification mechanism. To maintain data diversity consistent with LoDa's strategy (which resizes the short edge to 384 and then crops to 224), we adopt a random cropping ratio of $384/224 \approx 0.6$ relative to the shortest edge. Patches smaller than 224 are up-interpolated to meet the backbone's input requirements. We employ the AdamW optimizer with a learning rate of 1e-3 and weight decay of 0.01, using a batch size of 128. Unless otherwise specified, we utilize CLIP with the ViT-B/32 backbone as default. Additional Implementation details and settings are provided in the Appendix. C.

### 4.2. Performance Comparison with SOTA

We compare our CMPA fine-tuning framework against a wide range of state-of-the-art NR-IQA methods (Zhang et al., 2015; 2020; Zhu et al., 2020; Ying et al., 2021; Su et al., 2020; Ke et al., 2021; Zhang et al., 2023; Qin et al., 2023; Saha et al., 2023; Guan et al., 2024; Li et al., 2025; Xu et al., 2024). The results are summarized in Table 1. Our method consistently outperforms existing approaches across both synthetically distorted and authentically distorted datasets, achieving the highest or near-highest SRCC and PLCC in most cases. This substantial performance margin demonstrates the effectiveness of our proposed CMPA framework in capturing perceptual quality across diverse distortion types. Notably, our method achieves these results

*Table 1.* Performance comparison measured by medians of SRCC and PLCC, where the numbers within parentheses indicate the fine-tuned parameters of the model and **bold** entries indicate the top two results.

| Method | LIVE | | CSIQ | | TID2013 | | KADID-10k | | KonIQ-10k | | LIVEC | | SPAQ | | FLIVE | |
|---|---|---|---|---|---|---|---|---|---|---|---|---|---|---|---|---|
| | SRCC | PLCC | SRCC | PLCC | SRCC | PLCC | SRCC | PLCC | SRCC | PLCC | SRCC | PLCC | SRCC | PLCC | SRCC | PLCC |
| ILNIQE | 0.902 | 0.906 | 0.822 | 0.865 | 0.521 | 0.648 | 0.534 | 0.558 | 0.523 | 0.537 | 0.508 | 0.508 | 0.713 | 0.712 | 0.294 | 0.332 |
| DBCNN | 0.968 | 0.971 | 0.946 | 0.959 | 0.816 | 0.865 | 0.851 | 0.856 | 0.875 | 0.884 | 0.851 | 0.869 | 0.911 | 0.915 | 0.545 | 0.551 |
| MetaIQA | 0.960 | 0.959 | 0.899 | 0.908 | 0.856 | 0.868 | 0.762 | 0.775 | 0.887 | 0.856 | 0.835 | 0.802 | - | - | 0.540 | 0.507 |
| P2P-BM | 0.959 | 0.958 | 0.899 | 0.902 | 0.862 | 0.856 | 0.840 | 0.849 | 0.872 | 0.885 | 0.844 | 0.842 | - | - | 0.526 | 0.598 |
| HyperIQA(*27M*) | 0.962 | 0.966 | 0.923 | 0.942 | 0.840 | 0.858 | 0.852 | 0.845 | 0.906 | 0.917 | 0.859 | 0.882 | 0.911 | 0.915 | 0.544 | 0.602 |
| MUSIQ(*27M*) | 0.940 | 0.911 | 0.871 | 0.893 | 0.773 | 0.815 | 0.875 | 0.872 | 0.916 | 0.928 | 0.702 | 0.746 | 0.918 | 0.921 | 0.566 | 0.661 |
| TReS(*152M*) | 0.969 | 0.968 | 0.922 | 0.942 | 0.863 | 0.883 | 0.859 | 0.858 | 0.915 | 0.928 | 0.846 | 0.877 | - | - | 0.544 | 0.625 |
| LIQE(*151M*) | 0.970 | 0.951 | 0.943 | 0.946 | - | - | 0.930 | 0.931 | 0.919 | 0.908 | **0.904** | 0.910 | - | - | - | - |
| DEIQT(*24M*) | 0.980 | 0.982 | - | - | 0.892 | 0.908 | 0.889 | 0.887 | 0.921 | 0.934 | 0.875 | 0.894 | 0.919 | 0.923 | 0.571 | 0.663 |
| Re-IQA(*48M*) | 0.970 | 0.971 | 0.945 | 0.960 | 0.804 | 0.861 | 0.872 | 0.885 | 0.914 | 0.923 | 0.840 | 0.854 | 0.918 | 0.925 | 0.575 | 0.675 |
| QMamba(*50M*) | 0.959 | 0.958 | 0.950 | 0.952 | **0.896** | **0.913** | 0.923 | 0.938 | 0.928 | 0.943 | 0.863 | 0.903 | 0.927 | 0.933 | 0.574 | 0.672 |
| GRMP-IQA | 0.981 | 0.983 | 0.949 | 0.955 | - | - | - | - | 0.934 | 0.945 | 0.897 | **0.916** | 0.927 | **0.932** | **0.616** | **0.704** |
| LoDa(*9M*) | 0.975 | 0.979 | - | - | 0.869 | 0.901 | 0.931 | 0.936 | 0.932 | 0.944 | 0.876 | 0.899 | 0.925 | 0.928 | 0.578 | 0.679 |
| Ours(*w/o PAI 1.5M*) | 0.977 | 0.981 | 0.947 | 0.957 | 0.873 | 0.902 | 0.936 | 0.938 | 0.933 | 0.944 | 0.901 | 0.911 | 0.925 | 0.927 | 0.579 | 0.679 |
| Ours(*1.8M*) | **0.982** | **0.985** | **0.952** | **0.962** | 0.878 | 0.905 | **0.937** | **0.940** | **0.935** | **0.945** | 0.903 | 0.915 | **0.927** | 0.931 | 0.584 | 0.686 |
| Ours(with RPD) | **0.983** | **0.987** | 0.951 | 0.961 | **0.899** | **0.911** | **0.938** | **0.941** | **0.938** | **0.948** | 0.903 | 0.918 | 0.929 | 0.934 | 0.587 | 0.689 |

*Table 2.* SRCC on the cross datasets validation. The best performances are highlighted with **boldface**, and subsequent tables maintain the same.

| Training | FLIVE | | LIVEC | | KonIQ |
|---|---|---|---|---|---|
| Testing | KonIQ | LIVEC | KonIQ | LIVEC | |
| DBCNN | 0.716 | 0.724 | 0.754 | 0.755 | |
| P2P-BM | 0.755 | 0.738 | 0.740 | 0.770 | |
| HyperIQA | 0.758 | 0.735 | 0.772 | 0.785 | |
| TReS | 0.713 | 0.740 | 0.733 | 0.786 | |
| DEIQT | 0.733 | 0.781 | 0.744 | 0.794 | |
| LoDa | 0.763 | 0.805 | 0.745 | 0.811 | |
| Ours(w/o PAI) | **0.811** | **0.821** | **0.791** | **0.840** | |
| Ours | **0.825** | **0.820** | **0.801** | **0.846** | |

*Table 3.* Data-efficient learning validation with the training set containing 20%, 40% and 60% images.

| Mode | Methods | KonIQ | | LIVEC | |
|---|---|---|---|---|---|
| | | SRCC | PLCC | SRCC | PLCC |
| 20% | HyperIQA | 0.869 | 0.873 | 0.776 | 0.809 |
| | DEIQT | 0.888 | 0.908 | 0.792 | 0.822 |
| | LoDa | 0.907 | 0.923 | 0.815 | 0.854 |
| | Ours | **0.917** | **0.933** | **0.850** | **0.873** |
| 40% | HyperIQA | 0.892 | 0.908 | 0.832 | 0.849 |
| | DEIQT | 0.903 | 0.922 | 0.838 | 0.855 |
| | LoDa | 0.922 | 0.935 | 0.849 | 0.879 |
| | Ours | **0.926** | **0.941** | **0.885** | **0.906** |
| 60% | HyperIQA | 0.901 | 0.914 | 0.843 | 0.862 |
| | DEIQT | 0.914 | 0.931 | 0.848 | 0.877 |
| | LoDa | 0.928 | 0.940 | 0.869 | 0.891 |
| | Ours | **0.931** | **0.946** | **0.898** | **0.915** |

with highly parameter-efficient fine-tuning: only **1.8M** learnable parameters are tuned, far fewer than full fine-tuning of CLIP-based models or other parameter-heavy approaches. This highlights the parameter-friendliness of CMPA, making it practical for resource-constrained settings while delivering superior perceptual alignment. Compared to other CLIP-based methods (e.g., LIQE, GRMP-IQA), our approach benefits from explicit cross-modal guidance and refinement in the low-dimensional perceptual subspace. This design enables stronger perceptual sensitivity, leading to improved performance on both synthetic and authentic distortions.

### 4.3. Cross-Dataset Evaluation

We further compare the generalizability of our method against competitive BIQA models in a cross-dataset setting following (Qin et al., 2023). Training is performed on one specific dataset, and testing is conducted on a different dataset without any fine-tuning or parameter adaptation.

The experimental results, reported as the median SRCC across four datasets, are shown in Table 2. As observed, our method achieves the best performance on all datasets. Even without the further interactive guidance and refinement provided by PAI, our method can achieve strong capabilities simply through low-dimensional cross-modal alignment. These results manifest the generalization capability of the proposed method.

### 4.4. Data-Efficient Learning Validation

Following the evaluation protocol in DEIQT (Qin et al., 2023), we investigate the robustness of our fine-tuning framework under varying amounts of training data. We train on 20%, 40%, and 60% of the full training set from KonIQ-10k (randomly subsampled while preserving the

*Table 4.* Comparison of different CLIP fine-tuning strategies on KADID-10k, KonIQ-10k, and SPAQ (SRCC / PLCC). Best results in **bold**.

| Fine-tuning Method | KADID-10k | | KonIQ-10k | | SPAQ | |
|---|---|---|---|---|---|---|
| | SRCC | PLCC | SRCC | PLCC | SRCC | PLCC |
| CLIP (Original) | 0.666 | 0.671 | 0.625 | 0.727 | 0.738 | 0.735 |
| CLIP (Full fine-tune) | 0.869 | 0.879 | 0.928 | 0.894 | 0.898 | 0.902 |
| Adapter-CLIP (Gao et al., 2024) | 0.928 | 0.930 | 0.874 | 0.893 | 0.914 | 0.915 |
| LoRA (Hu et al., 2022) | 0.910 | 0.911 | 0.918 | 0.923 | 0.912 | 0.915 |
| COOP (Zhou et al., 2022) | 0.901 | 0.938 | 0.895 | 0.904 | 0.864 | 0.866 |
| Ours (w/o attn) | 0.936 | 0.933 | 0.933 | 0.945 | 0.925 | 0.927 |
| Ours | **0.937** | **0.940** | **0.935** | **0.945** | **0.927** | **0.931** |

original distribution), and evaluate on the standard test splits of KonIQ-10k and LIVEC using SRCC and PLCC. Our method consistently outperforms HyperIQA, DEIQT, and LoDa across all data regimes and both test sets. Notably, with only 40% of the training data, our approach achieves performance comparable to or better than the competitors' full-data results. Even at 20% data, we remain highly competitive, demonstrating strong data efficiency. These results highlight the effectiveness of our fine-tuning framework in low-data regimes, where perceptual alignment is preserved with minimal samples. This efficiency is particularly valuable for real-world scenarios with limited labeled perceptual data. Results are summarized in Table 3.

### 4.5. Comparison with Other Fine-Tuning Strategies

To fairly assess the effectiveness of our CMPA fine-tuning framework, we compare it against several established CLIP-based adaptation methods under identical experimental conditions: all models use the same random crop-based pre-input processing (without our RPD sampling). Results are reported in Table 4. Full-parameter fine-tuning of CLIP yields reasonable gains over zero-shot, but common parameter-efficient methods (LoRA, COOP) underperform or show only marginal improvement. This is likely because they operate directly in the original high-dimensional feature space, where semantic and perceptual signals are entangled, diluting the perceptual signal during adaptation. Adapter-CLIP achieves the second-best performance among baselines, thanks to its moderate dimensionality reduction (to half the original dimension) followed by up-projection. This partial dimensionality reduction inadvertently isolates some perceptual-sensitive components, providing partial support for our low-dimensional perceptual subspace hypothesis. Our CMPA framework outperforms all compared methods across all datasets. Ablating the PAI attention module (Ours w/o attn) already yields strong results, demonstrating the effectiveness of the perceptual feature enhancement (PFE) component alone. Adding the PAI module further boosts performance by guiding low-dimensional features toward perceptually sensitive directions through attention-based interaction and alignment. This incremental gain validates our core design: combining explicit perceptual subspace

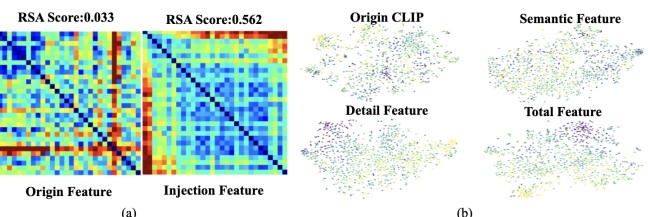

*Figure 4.* Visualization of representational alignment and feature distributions. (a) RDM results show that the injected features achieve a significantly higher correlation with human perception (0.562) compared to the original CLIP (0.033). (b) t-SNE visualizations demonstrate that while original CLIP is dominated by semantic clustering, our injected residuals and fused features exhibit perceptual discriminability and hierarchical structure.

enhancement with guided alignment yields the strongest perceptual fidelity. These results confirm that directly fine-tuning in the original CLIP space is suboptimal for perceptual tasks, while our low-dimensional, perceptually guided approach provides a more effective pathway.

### 4.6. Qualitative Analysis

To validate the hypothesis of a low-dimensional perceptual subspace, we visualize the feature distributions using RSA and t-SNE. As shown in Fig. 4(a), the original CLIP features exhibit negligible correlation with human perceptual judgments (RSA=0.033). In contrast, the injected perceptual residuals achieve a significantly higher RSA score of 0.562, confirming that our CMPA effectively isolates perceptually salient information from the semantic-dominant CLIP space. The t-SNE projections in Fig. 4(b) further reveal that while original CLIP features form rigid semantic clusters, the injected residuals exhibit a hierarchical structure sensitive to distortion levels and textures rather than high-level semantics. The fused "Total Feature" achieves superior perceptual discriminability by dispersing representations along perceptual axes while preserving necessary semantic coherence. These visualizations provide direct, qualitative support for our core hypothesis: perceptual information resides in a separable low-dimensional subspace within CLIP embeddings, and our CMPA framework—through guided cross-modal alignment—successfully extracts and enhances this subspace. The improved perceptual clustering in the fused features directly explains the superior performance observed in quantitative evaluations.

We further visualize the downsampled images produced by our RPD method alongside traditional interpolation baselines (Bicubic and Lanczos). The results, including zoomed-in regions with corresponding SSIM and LPIPS scores relative to the original high-resolution image, are shown in Figure 5. RPD consistently achieves higher SSIM and lower LPIPS scores compared to Bicubic and Lanczos, indicating superior structural preservation and perceptual fidelity.

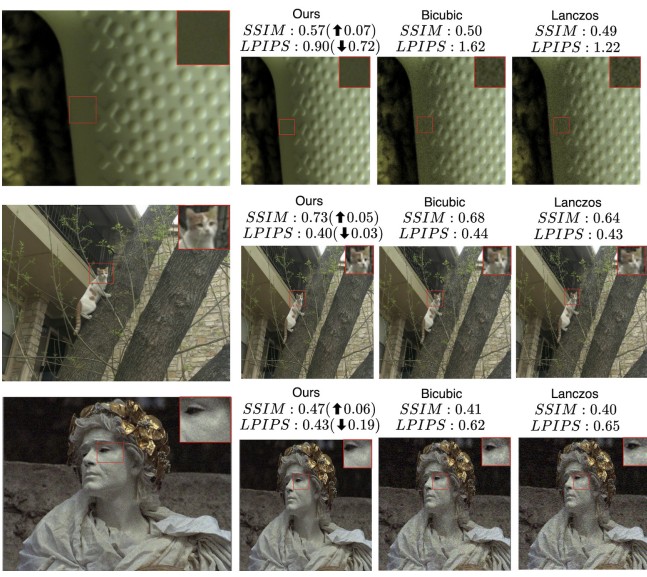

*Figure 5.* Visual comparison with other pre-input Downsampling methods. Zoom in for better.

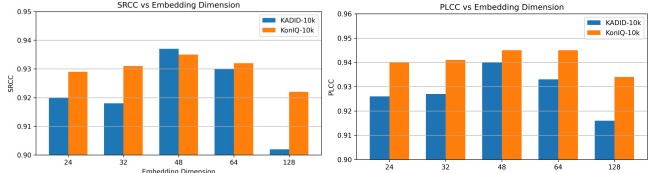

*Figure 6.* Impact of the perceptual subspace dimensionality. Performance across two datasets exhibits a non-monotonic trend, peaking at 48 dimensions before declining as the dimension increases towards 128. This inverted-U shape confirms that a compact low-dimensional space effectively isolates perceptual information from semantic entanglement.

*Table 5.* Performance gain when applied to HyperIQA and MANIQA.

| Method | LIVEC | | KonIQ-10k | | SPAQ | |
|---|---|---|---|---|---|---|
| | SRCC | PLCC | SRCC | PLCC | SRCC | PLCC |
| HyperIQA | 0.859 | 0.882 | 0.906 | 0.917 | 0.911 | 0.915 |
| + RPD (Ours) | **0.868** | **0.887** | **0.911** | **0.923** | **0.916** | **0.918** |
| MANIQA | 0.868 | 0.891 | 0.930 | 0.945 | 0.920 | 0.923 |
| + RPD (Ours) | **0.870** | **0.901** | **0.932** | **0.946** | **0.922** | **0.927** |

Visually, RPD outputs exhibit sharper details, reduced aliasing, and better texture retention, aligning more closely with human perception of the original image. For instance, in textured areas, RPD maintains fine-grained details that are blurred or lost in baselines. These qualitative comparisons demonstrate that RPD not only excels in objective metrics but also provides outputs with enhanced visual consistency to the originals. Additional visualizations are provided in the Appendix. E.

### 4.7. Ablation Studies

We conduct a series of ablation studies to validate the contribution. More ablation experiments can be found in the Appendix. D.

**Impact of Low-Dimensional Perceptual Subspace Dimension**: Following LoDa (Xu et al., 2024), we ablate the dimensionality of the perceptual subspace (24, 32, 48, 64, 128) while keeping other settings fixed. Performance is evaluated on KADID-10k and KonIQ-10k using SRCC and PLCC, as shown in Figure 6. The results reveal a non-monotonic trend: performance peaks at 48 dimensions on both datasets, then declines as dimensionality increases toward the original CLIP embedding size (128). This inverted-U shape is strikingly consistent with observations in LoDa and provides strong empirical support for our hypothesis: perceptual information is most salient and separable in a compact low-dimensional subspace. Excessive dimensionality reintroduces semantic entanglement, diluting perceptual sensitivity. More experiments related to the sensitivity of perception in low-dimensional spaces can be found in the Appendix. A.

**RPD Framework Generality Ablation** : To verify the generality of RPD beyond our CMPA framework, we apply it as a plug-and-play preprocessing step to two representative baselines: HyperIQA (CNN-based) and MANIQA (Transformer-based). Results are shown in Table 5. RPD consistently improves both baselines across all datasets. This framework-agnostic enhancement confirms that RPD's residual-enhanced perceptual downsampling is not tied to our specific CMPA architecture but provides general perceptual fidelity benefits when used as a preprocessing module. More reliability experiments related to RPD can be found in the Appendix. F.

## 5. Conclusion

In this paper, we address the discrepancy between CLIP's semantic invariance and the perceptual requirements of NR-IQA via the Cross-modal Perception Alignment Adapter (CMPA), which mitigates perceptual submergence by decoupling quality cues through low-dimensional manifold projection. This framework is further supported by Residual-enhanced Perceptual Downscaling (RPD) to preserve high-frequency fidelity during mandatory resizing to get perception-consistent input. Extensive benchmarks demonstrate that our approach offers a competitive, parameter-efficient solution, highlighting that manifold-aware decoupling and input fidelity are essential for adapting foundation models to fine-grained perceptual tasks. This work represents a meaningful step toward more robust and perception-aware VLM adaptation.

# 6. Additional Rebuttal Response

## 6.1. More Detailed Component Ablation Experiments

To further disentangle the source of performance gains, we conduct a fine-grained ablation on the bottleneck structure, cross-modal interaction, and attention design. As shown in Table 6, image-only tuning already provides the major improvement over the frozen CLIP baseline, while independently tuning both text and image branches brings almost no additional gain. In contrast, introducing the proposed Perceptual Feature Extraction (PFE) module with a shared low-dimensional subspace significantly improves performance, and the proposed Perceptual Attention Interaction (PAI) further provides consistent gains through explicit bidirectional cross-modal alignment. These results demonstrate that the improvements originate from the combination of compact perceptual subspace learning and active cross-modal guidance, rather than from generic bottleneck regularization alone.

*Table 6.* Detailed ablation of CMPA components on KADID-10k.

| Method | SRCC | PLCC |
|---|---|---|
| Frozen CLIP | 0.656 | 0.671 |
| + Image-only Adapter (high-d) | 0.929 | 0.932 |
| + Image-only Adapter (low-d) | 0.928 | 0.930 |
| + Independent Text+Image Tuning | 0.928 | 0.931 |
| + PFE (shared MLP, high-d) | 0.912 | 0.916 |
| + PFE (shared MLP, low-d) | 0.936 | 0.938 |
| + PAI (text→image) | 0.938 | 0.941 |
| + PAI (Full CMPA) | **0.938** | **0.941** |

## 6.2. Hyperparameter Sensitivity Analysis

We further analyze the sensitivity of the RPD hyperparameters $(\alpha, \beta, \eta)$ by varying each parameter within $\{0.5, 0.8, 1.0, 1.2, 1.5\}$ while fixing the remaining two. As shown in Table 7, the performance remains highly stable across a broad range, demonstrating that RPD does not rely on delicate hyperparameter tuning.

*Table 7.* Hyperparameter sensitivity analysis of RPD on KonIQ-10k (SRCC/SSIM).

| Value | $\alpha$ | $\beta$ | $\eta$ |
|---|---|---|---|
| 0.5 | 0.932/0.860 | 0.934/0.851 | 0.930/0.859 |
| 0.8 | 0.934/0.874 | 0.935/0.869 | 0.933/0.865 |
| 1.0 | **0.938**/0.871 | **0.938**/0.871 | **0.938**/0.871 |
| 1.2 | 0.935/0.857 | 0.936/0.861 | 0.935/0.851 |
| 1.5 | 0.930/0.865 | 0.931/0.832 | 0.929/0.821 |

## 6.3. More Detailed Analysis of Time Consumption

We provide a detailed efficiency analysis including sampling latency, inference time, GPU memory footprint, GFLOPs, and parameter count in Table 8. The proposed CMPA introduces only 1.8M trainable parameters ($< 2\%$ of CLIP ViT-B), while RPD itself is entirely parameter-free. Although RPD mathematically requires two interpolation passes, the practical overhead remains extremely small compared with the backbone computation. Specifically, RPD increases sampling latency from only 0.005s to 0.011s, while inference latency remains nearly unchanged (1.08 ms/image). These results demonstrate that the proposed framework preserves strong practicality for large-scale training and real-time deployment.

*Table 8.* Detailed efficiency comparison (batch size = 128).

| Method | Samp. (s) | Infer. (ms/img) | Mem. (MB) | GFLOPs | Total Params | Train. Params |
|---|---|---|---|---|---|---|
| CLIP (Resize) | 0.005 | 0.99 | 1652 | 27.29 | 151.5M | – |
| CMPA only | 0.005 | 1.08 | 2192 | 27.46 | 153.3M | 1.8M |
| RPD + CMPA | 0.011 | 1.08 | 2192 | 27.46 | 153.3M | 1.8M |

## 6.4. Transferability and CMPA Non-Regularizer Analysis

To validate the generalizability of CMPA, we extend experiments to additional VLM backbones including BLIP, SigLIP, and ALIGN. As shown in Table 9, CMPA consistently achieves strong performance on BLIP and SigLIP, demonstrating that the proposed low-dimensional perceptual decoupling generalizes well across modern high-quality multimodal encoders. Interestingly, the gains on ALIGN are significantly weaker. We argue that this phenomenon provides evidence that CMPA is not merely a generic regularizer. BLIP and SigLIP are trained on carefully filtered or curated data, resulting in well-structured latent spaces where coherent perceptual manifolds exist and can be effectively isolated by CMPA. In contrast, ALIGN is trained on extremely noisy web-scale image-text pairs, causing perceptual cues to be heavily corrupted. Consequently, CMPA cannot effectively recover a stable perceptual manifold from such noisy representations. This contrast strongly supports our core hypothesis that CMPA specifically unlocks structured perceptual representations rather than acting as a generic capacity-reduction module.

*Table 9.* Transferability of CMPA across different VLM backbones (SRCC).

| Method | KADID-10k | KonIQ-10k | LIVEC |
|---|---|---|---|
| ALIGN | 0.910 | 0.907 | 0.838 |
| BLIP | 0.936 | 0.936 | 0.895 |
| SigLIP | 0.933 | 0.937 | 0.897 |

## Impact Statement

This paper presents work whose goal is to advance the field of Machine Learning. There are many potential societal consequences of our work, none which we feel must be specifically highlighted here.

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

## A. Statistical Validation of the Subspace Distortion Amplification Effect

### A.1. Dimensionality Reduction Toy Experiment

One might hypothesize that the performance gain comes from reduced overfitting due to fewer parameters. However, our RDM analysis (Figure. 4a) reveals that the correlation with human perception increases significantly in the subspace, confirming that the gain stems from semantic disentanglement rather than mere regularization. To further validate our hypothesis of a low-dimensional perceptual subspace within CLIP embeddings, we perform an additional dimensionality reduction study. Features are extracted using a frozen CLIP image encoder. PCA is applied to project these into a lower-dimensional subspace, after which the reduced features are interpolated back to the original dimensionality. The residual between this interpolated representation and the original features constitutes our perceptual feature, consistent with the residual injection strategy in CMPA.

We conduct a controlled toy experiment comparing distortion sensitivity between the original high-dimensional features and the residual subspace. We sample distorted images from the KADID-10k (Lin et al., 2019). Deviation intensity (normalized displacement from pristine) and Signal-to-Noise Ratio (SNR) are computed for both representations across distortion levels.

As shown in Figure 7, the residual subspace consistently exhibits higher distortion sensitivity, with SNR gains up to **1.59×** at subtle distortion levels and ranging from **1.24×** to **1.51×** overall. Box plots of deviation intensity further reveal amplified responses, particularly to mild distortions. These findings demonstrate that linear dimensionality reduction can isolate and enhance perceptually salient components, providing strong evidence for the viability of our low-dimensional perceptual subspace hypothesis and its potential utility in perceptual alignment tasks.

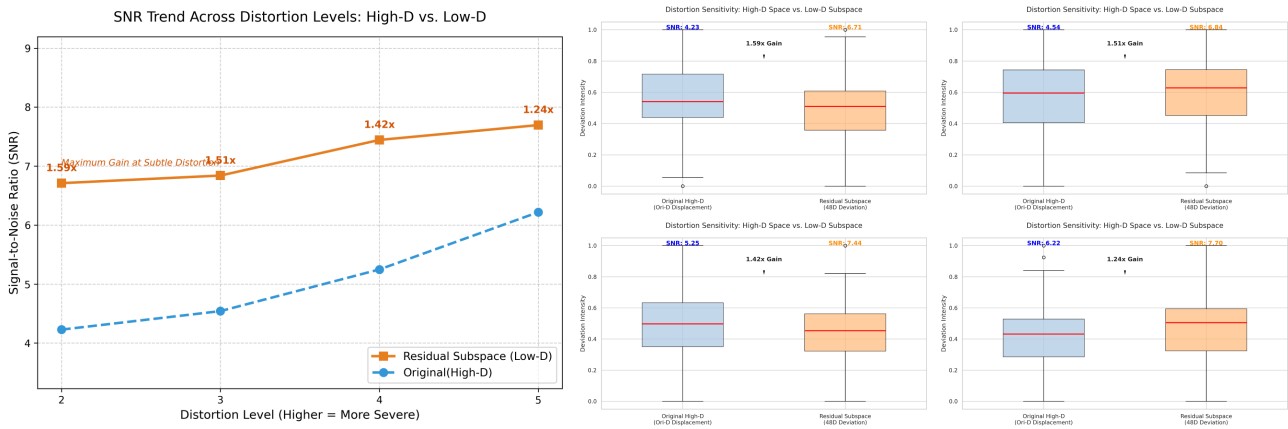

*Figure 7.* Distortion sensitivity analysis in high-dimensional vs. low-dimensional spaces. Our residual subspace consistently yields higher Signal-to-Noise Ratio (SNR) and amplified deviation intensity across all distortion levels compared to the original CLIP space. The significant sensitivity gain (up to 1.59×) at subtle distortions validates that a compact subspace effectively isolates and enhances perceptually salient information.

## B. Optimization-inspired Theoretical Analysis of RPD

We present a heuristic framework viewing the proposed downsampling method as a first-order approximation to a perceptually guided convex optimization problem, capturing the essence of Human Visual System (HVS), driven error minimization while opening doors for further refinement.

### B.1. Problem Formulation

Let $\mathbf{x} \in \mathbb{R}^N$ be the high-resolution input and $\mathbf{y} \in \mathbb{R}^M$ ($M < N$) the target low-resolution output. With linear downsampling $\mathcal{D}$ and upsampling $\mathcal{U}$, perceptual downsampling seeks $\mathbf{y}^*$ minimizing the HVS-weighted reconstruction error:

$$\min_{\mathbf{y}} f(\mathbf{y}) = \frac{1}{2}\|\mathbf{W}(\mathcal{U}\mathbf{y} - \mathbf{x})\|_2^2 \tag{10}$$

where $\mathbf{W} = \mathrm{diag}(w_i)$ encodes NSS-derived JND weights reflecting local luminance adaptation and texture masking.

## B.2. Adaptive Gain via Weber-Fechner Law

The Weber-Fechner law describes the relationship between the physical intensity of a stimulus $I$ and its perceived magnitude $S$. In its classical form, it states that the perceived change in sensation is proportional to the relative change in stimulus intensity:

$$\Delta S = k \cdot \frac{\Delta I}{I} \quad \text{or, in integrated form,} \quad S = k \ln I + C, \tag{11}$$

where $k$ is a constant and $C$ is an integration constant. This implies that human perception follows a logarithmic response: larger absolute changes are required to produce the same perceived difference at higher stimulus levels (diminishing marginal sensitivity).

In the context of perceptual downsampling, we introduce an adaptive gain $G$ to compensate for information loss while respecting this perceptual nonlinearity. When the downsampling ratio $d = N/M$ increases, more high-frequency content is discarded, but human observers do not perceive the loss linearly. Instead, the need for compensation grows sub-linearly with $d$.

We therefore define the gain as

$$G(d, \mathbf{W}) = \mathbf{1} + \eta \cdot \log_2 \left( 1 + \left( 1 - \frac{1}{d} \right) \right) \cdot \mathbf{W}, \tag{12}$$

where $\eta > 0$ is a scaling factor controlling the strength of enhancement, and $\mathbf{W}$ incorporates local JND-based weights. The logarithmic term $\log_2(1 + (1 - 1/d))$ reflects Weber-Fechner's principle: as $d$ grows larger, additional compensation yields progressively smaller perceptual benefit, naturally preventing over-sharpening or halo artifacts in regions where further enhancement would be less noticeable.

This formulation ensures that the residual injection is perceptually economical — stronger enhancement is applied only where it is most likely to be noticed (guided by $\mathbf{W}$) and scales sub-linearly with the degree of downsampling, aligning the method with human visual sensitivity.

## B.3. First-Order Approximation to the Optimum

The convex objective (1) has gradient $\nabla_{\mathbf{y}} f = \mathcal{U}^\top \mathbf{W}^2 (\mathcal{U}\mathbf{y} - \mathbf{x})$. Starting from $\mathbf{y}_0 = \mathcal{D}\mathbf{x}$, one-step gradient descent yields:

$$\mathbf{y}_1 = \mathcal{D}\mathbf{x} + \alpha \mathcal{U}^\top \mathbf{W}^2 (\mathbf{x} - \mathcal{U}\mathcal{D}\mathbf{x}) \tag{13}$$

Approximating $\alpha \mathbf{W}^2 \approx G(d, \mathbf{W})$ (with $\mathbf{W}^2 \approx c\mathbf{W}$ under small/normalized $\mathbf{W}$) and $\mathcal{U}^\top \approx \mathcal{D}$, we obtain the practical form:

$$\mathbf{y}_{\text{final}} \approx \mathcal{D}\mathbf{x} + G(d, \mathbf{W}) \odot \mathcal{D}(\mathbf{R}) \tag{14}$$

where $\mathbf{R} = \mathbf{x} - \mathcal{U}\mathcal{D}\mathbf{x}$ is the lost high-frequency residual. This closed-form residual injection approximates perceptual error minimization, with bounded approximation error $O(\alpha^2 \|\mathbf{R}\|^2)$ under mild conditions.

## B.4. Information-Theoretic Insight

Standard downsampling discards frequencies above the Nyquist limit. The residual enhancement selectively projects perceptually salient high frequencies back into the low-resolution domain, guided by JND thresholds. This maximizes perceptual mutual information $I_{\text{HVS}}(\mathbf{x}; \mathbf{y})$ under the resolution constraint, echoing perceptual losses in modern super-resolution (e.g., ESRGAN (Wang et al., 2018)) and inspiring extensions to deep learning, compression, and content-aware processing.

This heuristic framework bridges classical optimization and HVS principles, offering a foundation for perceptually intelligent image algorithms with room for rigorous refinement and broader impact.

## C. Additional implementation details

**More Dataset Information** : For the synthetic datasets, they contain a few pristine im-ages that are synthetically distorted by various distortion types. LIVE (Sheikh et al., 2006) contains 779 synthetically distorted images with 5 distortion types. The CSIQ (Larson & Chandler, 2010) dataset is a synthetic IQA benchmark containing 30 reference images and 866 distorted images with various distortion types and subjective quality scores. TID2013 (Ponomarenko et al., 2015)andKADID-10k (Lin et al., 2019) consist of 3,000 and 10,125 synthetically distorted images involving 24 and 25 distortion types, respectively. For the authentic datasets, LIVEC (Ghadiyaram & Bovik, 2015) consists of 1,162 images with diverse authentic distortions captured by mobile devices. KonIQ10k (Hosu et al., 2020) contains 10,073 images which are selected from YFCC100M and the selected images cover a wide and uniform range of distortions such as brightness colorfulness, contrast, noise, sharpness, etc. SPAQ (Fang et al., 2020) consists of 11,125 images captured by different mobile devices, covering a large variety of scene categories. FLIVE (Ying et al., 2021) is the largest in-the-wild IQA dataset by far, which contains 39,810 real-world images with diverse contents, sizes, and aspect ratios.

**Training and Evaluation Specifics**: To ensure the sampling process captures comprehensive image information, random horizontal and vertical flips are applied during training. We adopt a standard 80/20 train-test split ratio. To minimize performance bias and ensure statistical robustness, each experiment is repeated 10 times, and the median Spearman Rank-Order Correlation Coefficient (SRCC) and Pearson Linear Correlation Coefficient (PLCC) are reported. All models are trained and evaluated on a single NVIDIA RTX 4090 GPU. $\alpha, \beta$ are all set to 1.0 in the experiments. The $\sigma$ of the Gaussian kernel separated in the frequency domain is inspired by (Arslan et al., 2025) and designed to be 0.5 times the scaling ratio $d$.

**Loss Function**: For the loss function, we use Pearson Linear Correlation Coefficient (PLCC) loss (Li et al., 2020), which can be formally expressed as

$$L_{\text{PLCC}} = 1 - \frac{\sum_{i=1}^{m} \left( \tilde{y}_i - \tilde{a} \right) \left( y_i - a \right)}{\sqrt{\sum_{i=1}^{m} \left( \tilde{y}_i - \tilde{a} \right)^2 \sum_{i=1}^{m} \left( y_i - a \right)^2}}) \quad (15)$$

where $\tilde{y}_i$ is the predicted quality score for the $i$-th image, $y_i$ is the true quality score for the $i$-th image, $\tilde{a} = \frac{1}{m} \sum_{i=1}^{m} \tilde{y}_i$ and $a = \frac{1}{m} \sum_{i=1}^{m} y_i$ are the mean values of the predicted and true quality scores, respectively, $m$ is the number of images in the training batch.

## D. Additional ablation experiments

### D.1. Study on the scale of pre-trained CLIP

We further examine how the pre-training scale of the CLIP backbone affects CMPA performance. We compare three ViT variants: ViT-L/14, ViT-B/16, and ViT-B/32. Performance scales with backbone capacity, with ViT-L/14 achieving the

*Table 10.* Impact of large-scale pretrained model sizes.

| Backbone | KADID-10k | | KonIQ-10k | | SPAQ | |
|---|---|---|---|---|---|---|
| | SRCC | PLCC | SRCC | PLCC | SRCC | PLCC |
| ViT-L-14 | **0.951** | **0.953** | **0.936** | 0.945 | **0.929** | **0.932** |
| ViT-B-16 | 0.934 | 0.935 | 0.934 | 0.943 | 0.925 | 0.929 |
| ViT-B-32 | 0.938 | 0.941 | 0.935 | **0.945** | 0.927 | 0.931 |

strongest results on synthetic distortions (KADID-10k) and remaining highly competitive on authentic datasets. Notably, ViT-B/16 yields near-top performance across the board, demonstrating that our fine-tuning approach efficiently harnesses mid-scale pre-trained representations without excessive computational cost. These findings validate the effectiveness of CMPA and its ability to benefit from richer pre-training in perceptual quality assessment tasks.

### D.2. Performance on Individual Distortion Types

To verify that our Residual-Enhanced Perceptual Downsampling (RPD) does not cause over-enhancement for specific distortions, we evaluate per-distortion performance on CSIQ and LIVE datasets using SRCC and PLCC. Results are shown in Table 11. Our method outperforms MANIQA on most distortion types across both datasets (e.g., superior on JPEG, WN, GB in both; JP2K in CSIQ and LIVE). This demonstrates the effectiveness and broad robustness of RPD.

On Additive Pink Gaussian Noise (FN) in CSIQ, our performance is slightly lower than MANIQA. This is expected: RPD injects high-frequency residual compensation, which partially offsets the natural high-frequency attenuation of FN, which emphasizes more low-frequency structures, while the high-frequency components are relatively suppressed. However, JND-guided weighting limits the influence of these residuals, resulting in only negligible overall difference.

These results confirm that RPD enhances perceptual sensitivity without introducing systematic bias or significant degradation for any major distortion type.

*Table 11.* Performance comparison on LIVE and CSIQ datasets on Individual Distortion Types (PLCC).

| Dataset | LIVE | | | | | CSIQ | | | | | |
|---|---|---|---|---|---|---|---|---|---|---|---|
| | JP2K | JPEG | WN | GB | FF | WN | JPEG | JP2K | FN | GB | CC |
| BRISQUE | 0.929 | 0.965 | 0.982 | 0.964 | 0.828 | 0.723 | 0.806 | 0.840 | 0.378 | 0.820 | 0.804 |
| ILNIQE | 0.894 | 0.941 | 0.981 | 0.915 | 0.833 | 0.850 | 0.899 | 0.906 | 0.874 | 0.858 | 0.501 |
| HOSA | 0.935 | 0.954 | 0.975 | 0.954 | 0.954 | 0.604 | 0.733 | 0.818 | 0.500 | 0.841 | 0.716 |
| BIECON | 0.952 | **0.974** | 0.980 | 0.956 | 0.923 | 0.902 | 0.942 | 0.954 | 0.884 | 0.946 | 0.523 |
| WaDIQaM | 0.942 | 0.953 | 0.982 | 0.938 | 0.923 | 0.974 | 0.853 | 0.947 | 0.882 | 0.976 | 0.923 |
| PQR | **0.953** | 0.965 | 0.981 | 0.944 | 0.921 | 0.915 | 0.934 | 0.955 | 0.926 | 0.921 | 0.837 |
| HyperIQA | 0.949 | 0.961 | 0.982 | 0.926 | 0.934 | 0.927 | 0.934 | 0.960 | 0.931 | 0.915 | 0.874 |
| MANIQA | 0.870 | 0.895 | 0.984 | 0.959 | 0.896 | 0.966 | 0.971 | 0.973 | **0.977** | 0.956 | 0.946 |
| **Ours** | **0.988** | **0.990** | **0.984** | **0.983** | **0.971** | **0.984** | **0.982** | **0.986** | 0.973 | **0.976** | **0.949** |

### D.3. Study on Sampling Time and Performance Trade-off

To quantify the additional overhead introduced by our Residual-Enhanced Perceptual Downsampling (RPD), we measure sampling time and perceptual quality on the KonIQ-10k dataset using SSIM and LPIPS (higher is better for SSIM, while lower is better for LPIPS). Results are reported in Table 12 for single-step RPD (**Ours**), iterative RPD (**Ours_iter**), and baselines Lanczos and Bicubic.

*Table 12.* Sampling time (seconds) , perceptual quality (SSIM / LPIPS) and IQA metrics (PLCC/SRCC) on KonIQ-10k. Best perceptual scores in **bold**.

| Method | Sampling Time (s) | SSIM | LPIPS | SRCC | PLCC |
|---|---|---|---|---|---|
| Ours_iter | 0.053 | 0.821 | 0.310 | 0.910 | 0.924 |
| Ours | 0.011 | **0.871** | **0.241** | **0.929** | **0.945** |
| Lanczos | 0.007 | 0.852 | 0.265 | 0.914 | 0.931 |
| Bicubic | **0.005** | 0.854 | 0.265 | 0.921 | 0.935 |

Our single-step RPD incurs a modest time overhead ($\sim$2–3$\times$ over traditional interpolation) but delivers superior perceptual quality across both metrics. This pre-input sampling strategy adds negligible impact to model parameters or GFLOPs, as it occurs before the network forward pass. Iterative application of RPD (**Ours_iter**) further increases time cost and degrades performance due to excessive high-frequency enhancement, which disrupts perceptual structure and information fidelity. This highlights a clear trade-off: RPD provides strong perceptual gains at low single-step cost, but benefits most from non-iterative use to avoid over-compensation.

### D.4. Full Component Ablation

Although Table 1 has some ablation experimental results, further explanations are given here in order to avoid ambiguity. Table 13 compares the performance of the original CLIP (zero-shot), CLIP with CMPA fine-tuning, and the full framework with both CMPA and RPD. Evaluations are performed on KADID-10k, KonIQ-10k, and SPAQ using SRCC and PLCC. CLIP alone exhibits limited perceptual alignment in zero-shot settings. Applying CMPA fine-tuning yields substantial improvements across all datasets, demonstrating the effectiveness of our perceptual subspace enhancement and cross-modal

alignment. Adding RPD further boosts performance, particularly on authentic distortion benchmarks (e.g., KonIQ-10k SRCC from 0.935 to 0.938, SPAQ PLCC from 0.931 to 0.934). These incremental gains confirm that both CMPA and RPD contribute meaningfully, with RPD compensating for high-frequency information loss during downsampling.

*Table 13.* Full component ablation: Impact of CMPA and RPD on perceptual quality assessment.

| Backbone | KADID-10k | | KonIQ-10k | | SPAQ | |
|---|---|---|---|---|---|---|
| | SRCC | PLCC | SRCC | PLCC | SRCC | PLCC |
| CLIP (Original) | 0.656 | 0.671 | 0.625 | 0.727 | 0.738 | 0.735 |
| +CMPA | 0.937 | 0.940 | 0.935 | 0.945 | 0.927 | 0.931 |
| +CMPA & RPD | **0.938** | **0.941** | **0.938** | **0.948** | **0.929** | **0.934** |

### D.5. Component Ablation of CMPA

To gain deeper insight into how our CMPA framework operates, we perform a detailed component ablation study on its key modules: text-only fine-tuning, image-only fine-tuning, dual text-image fine-tuning, addition of the Perceptual Feature Enhancement (PFE) module with shared MLP, and the full CMPA with Perceptual Alignment Interaction (PAI) module. Results are summarized in Table 14. Fine-tuning only the text encoder yields moderate gains over zero-shot CLIP, but

*Table 14.* Impact of text-only, image-only, PFE, and PAI modules.

| Method | KADID-10k | | KonIQ-10k | | SPAQ | |
|---|---|---|---|---|---|---|
| | SRCC | PLCC | SRCC | PLCC | SRCC | PLCC |
| CLIP (Original) | 0.656 | 0.671 | 0.625 | 0.727 | 0.738 | 0.735 |
| + Only Text | 0.792 | 0.793 | 0.796 | 0.873 | 0.893 | 0.886 |
| + Only Image | 0.928 | 0.930 | 0.928 | 0.933 | 0.914 | 0.923 |
| + Text & Image | 0.928 | 0.931 | 0.920 | 0.934 | 0.914 | 0.920 |
| + PFE (shared MLP) | 0.936 | 0.938 | 0.933 | 0.945 | 0.925 | 0.927 |
| + PAI (Full CMPA) | **0.938** | **0.941** | **0.935** | **0.945** | **0.927** | **0.931** |

performance remains limited. This is because the image features remain in the original high-dimensional space, where perceptual signals are diluted by semantic information, leading to low perceptual sensitivity. Image-only fine-tuning achieves results comparable to CLIP-Adapter, as both operate primarily on the visual branch without strong cross-modal guidance. Adding the PFE module (shared MLP with perception-aware text prompts) forces image features toward the perceptual subspace, delivering a clear performance boost (e.g., SRCC from 0.928 to 0.936 on KADID-10k). This confirms the effectiveness of explicit perceptual enhancement. Incorporating the PAI module provides further improvement through attention-based cross-modal interaction and alignment, yielding the best results across all datasets. The incremental gain from PAI validates our design: guided interaction in the low-dimensional perceptual subspace refines alignment and enhances perceptual discriminability.

These ablations demonstrate that the full CMPA framework—combining perception-aware text guidance, feature enhancement (PFE), and cross-modal interaction (PAI), is essential for achieving superior perceptual quality assessment performance.

### D.6. Component Ablation of RPD

We performed ablation on the components of the RPD sampling, mainly eliminating the adaptive perceptual JND weights based on the scaling coefficient. The specific results are shown in the Table 15. It can be seen that without the adaptive weighting supplement method, although it improves the information retention of the sampling to a certain extent and enhances the perceptual performance on the real dataset, the perceptual performance on synthetic datasets such as Tid2013 remains poor. This is probably due to the overfitting caused by directly fusing the lost high-frequency information. With the addition of JND, the effect has been improved to some extent. Moreover, with the addition of perceptual weights, the effect has become even better, which indicates the importance of integrating human visual system and perceptual laws.

*Table 15.* Result on component ablation of RPD.

| Methods | KonIQ | | Tid2013 | |
|---|---|---|---|---|
| | SRCC | PLCC | SRCC | PLCC |
| Bicubic | 0.926 | 0.936 | 0.879 | 0.899 |
| + Residual information | 0.932 | 0.943 | 0.870 | 0.879 |
| + JND weight map | 0.933 | 0.945 | 0.883 | 0.907 |
| + Perceptual weight | **0.938** | **0.948** | **0.899** | **0.911** |

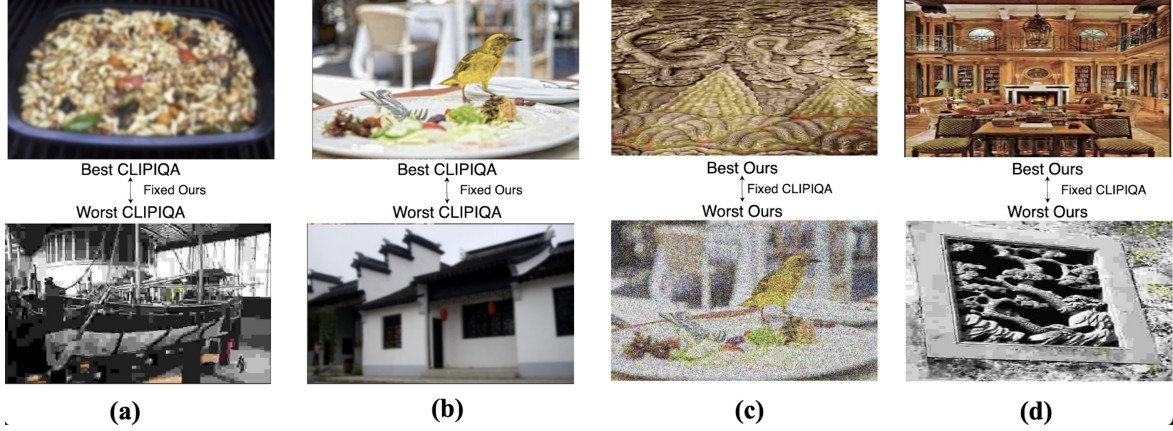

*Figure 8.* Representative gMAD pairs between proposed methods and CLIPIQA+ on synthetic distortions. From left to right: Fixing Ours at the low-quality. Fixing Ours at the high-quality. Fixing CLIPIQA+ at the low-quality. Fixing CLIPIQA+ at the high-quality.

## E. More Qualitative experiments

### E.1. More Qualitative gMAD result

To illustrate perceptual superiority, we conduct a qualitative gMAD competition against CLIPIQA+. The results are shown in Figure 8. Our method consistently produces downsampled images that are more perceptually distinguishable from each other in pairwise human-like judgments, outperforming CLIPIQA+ by a clear margin. This qualitative evidence reinforces the robustness and effectiveness of our fine-tuning strategy in generating perceptually faithful representations.

To further illustrate the perceptual superiority of our Residual-Enhanced Perceptual Downsampling (RPD), we conducted a qualitative GMAD (Ma et al., 2016) competition between RPD and standard Bicubic downsampling. GMAD evaluates how well different methods produce images that are perceptually distinguishable by human observers.

The results are shown in Figure 9. Our method consistently ranks higher in the GMAD pairwise comparison, generating downsampled images that better preserve perceptual details and overall quality compared to Bicubic. These qualitative findings confirm the stronger perceptual fidelity of RPD.

Additionally, we visualize the Fourier spectrum of the pre-input downsampled images produced by RPD, Lanczos, Bicubic, and the original high-resolution image (averaged over 128 images from SPAQ). The results are presented in Figure 10.

As shown in the Fourier spectra and relative log amplitudes of the transformed feature maps, RPD downsampled images retain significantly more high-frequency information than traditional interpolation methods (Lanczos and Bicubic). Moreover, the relative log amplitude distribution of RPD is much closer to that of the original high-resolution image. This indicates that RPD effectively compensates for high-frequency loss during downsampling, resulting in a more faithful information distribution that aligns better with the original content. Together, these qualitative and frequency-domain analyses provide compelling evidence of the perceptual advantages and robustness of the proposed RPD method over conventional downsampling techniques.

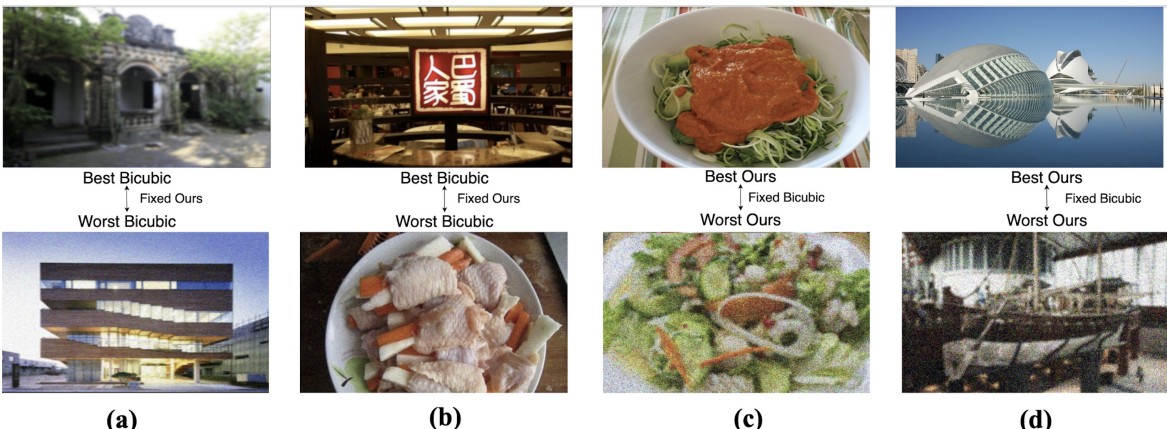

*Figure 9.* Representative gMAD pairs between proposed methods and Bicubic on synthetic distortions. From left to right: Fixing Ours at the low-quality. Fixing Ours at the high-quality. Fixing Bicubic at the low-quality. Fixing Bicubic at the high-quality.

### E.2. Qualitative Analysis of Saliency Maps

To qualitatively assess the perceptual focus of our CMPA framework, we visualize saliency maps for the original images, full-fine-tuned CLIP, CLIP-Adapter, our CMPA method, and CMPA with downsampling (using RPD). The saliency maps highlight regions of high activation, revealing where each method attends during perceptual quality assessment. Results are shown in Figure 11.

In the full-fine-tuned CLIP and CLIP-Adapter variants, saliency often concentrates on semantically dominant regions (e.g., central objects or high-contrast edges), with scattered activations that may overlook subtle perceptual distortions like texture degradation or noise. This suggests a bias toward semantic features rather than fine-grained perceptual details. In contrast, our CMPA method produces more focused and perceptually relevant saliency: activations are stronger in areas sensitive to human perception, such as textured surfaces, edges with potential aliasing, or regions prone to luminance variations. When combined with RPD downsampling, the saliency maps exhibit even greater refinement, with enhanced emphasis on high-frequency details that are typically lost in standard downsampling. This results in more accurate highlighting of perceptual artifacts (e.g., blurring in backgrounds or color shifts in complex scenes).

These qualitative patterns demonstrate that CMPA, especially with RPD, shifts attention toward perceptually salient features, aligning better with human visual sensitivity. The visualizations further validate the effectiveness of our low-dimensional perceptual subspace guidance and residual enhancement in producing robust, perception-aware representations.

## F. Rationality analysis of RPD

### F.1. Generalization of RPD without Additional Data Augmentation

**In-Dataset Evaluation**: To further demonstrate the intrinsic perceptual preservation capability of our Residual-Enhanced Perceptual Downsampling (RPD) and rule out potential confounding effects from other data augmentation strategies, we conduct an ablation study using **RPD alone** for downsampling during training—without any additional augmentations such as random cropping or flipping. All patches are of size 1 (full-image level), and other training settings remain identical to the main experiments. Results are shown in Table 16.

*Table 16.* Performance with RPD-only downsampling (no random crop/flip) vs. traditional downscaling methods.

| Method | LIVE | | KonIQ-10k | | SPAQ | | FLIVE | |
|---|---|---|---|---|---|---|---|---|
| | SRCC | PLCC | SRCC | PLCC | SRCC | PLCC | SRCC | PLCC |
| Bicubic | 0.863 | 0.882 | 0.921 | 0.935 | 0.915 | 0.918 | 0.561 | 0.673 |
| Lanczos | 0.851 | 0.863 | 0.914 | 0.931 | 0.901 | 0.907 | 0.545 | 0.665 |
| Ours (No-Aug) | **0.877** | **0.893** | **0.929** | **0.945** | **0.923** | **0.928** | **0.585** | **0.685** |

| **Lanczos** | **Bicubic** | **Ours** |
|:---:|:---:|:---:|

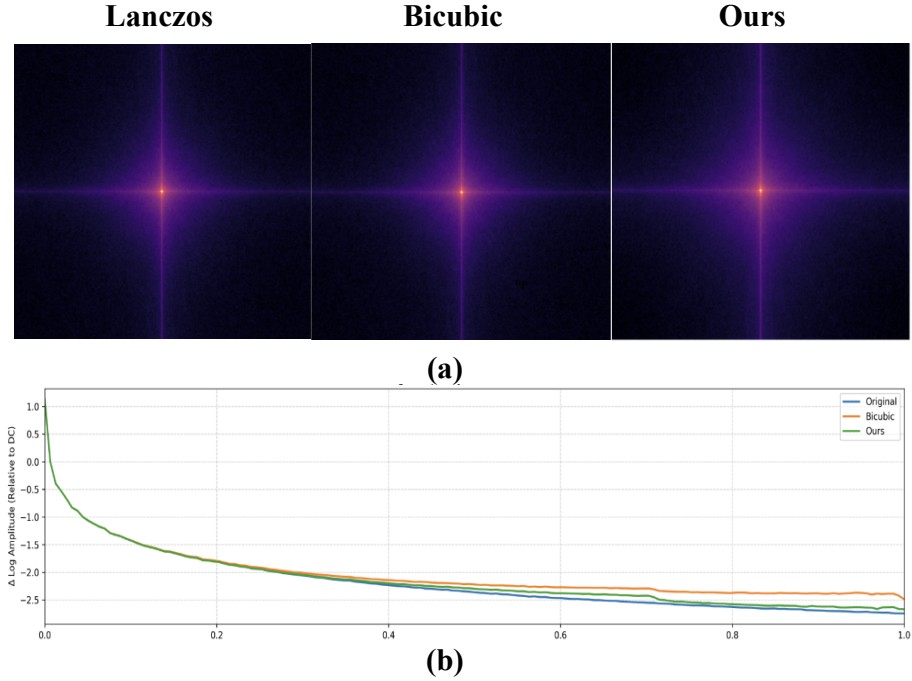

(a)

(b)

*Figure 10.* Fourier analysis was performed on the downsampled images obtained by the conventional interpolation and RPD methods. (a) Fourier spectrum of all methods. (b) Relative log amplitude of the Fourier transform feature map. (a) and (b) show that RPD captures more high-frequency signals while the frequency energy distribution is closer to the original image

Even without any conventional data augmentation to increase training diversity, our RPD-based training achieves competitive or near state-of-the-art performance across all datasets. This substantial improvement over Bicubic and Lanczos underscores that RPD's perceptual preservation is a fundamental property of the downsampling process itself, rather than an artifact of augmentation-induced diversity.

**Cross-Dataset Evaluation**: To rule out the possibility that this strong performance stems from dataset overfitting rather than genuine perceptual learning, we further perform cross-dataset evaluation. We train on one dataset and test on another, still using RPD-only downsampling without augmentation. Results are reported in Table 17.

*Table 17.* Cross-dataset generalization (SRCC) without data augmentation.

| Training | FLIVE | | LIVEC | KonIQ |
|:---:|:---:|:---:|:---:|:---:|
| Testing | KonIQ | LIVEC | KonIQ | LIVEC |
| DBCNN | 0.716 | 0.724 | 0.754 | 0.755 |
| P2P-BM | 0.755 | 0.738 | 0.740 | 0.770 |
| HyperIQA | 0.758 | 0.735 | 0.772 | 0.785 |
| TReS | 0.713 | 0.740 | 0.733 | 0.786 |
| DEIQT | 0.733 | 0.781 | 0.744 | 0.794 |
| LoDa | 0.763 | **0.805** | 0.745 | 0.811 |
| Ours (No-Aug) | **0.781** | 0.793 | **0.766** | **0.828** |

Despite the absence of augmentation-induced diversity, our method still achieves stronger cross-domain generalization than LoDa and remains competitive with augmentation-equipped baselines. This confirms that RPD enables the model to learn genuine perceptual representations rather than merely overfitting to training distribution artifacts. These ablation results highlight the robustness and effectiveness of RPD as a standalone perceptual downsampling strategy.

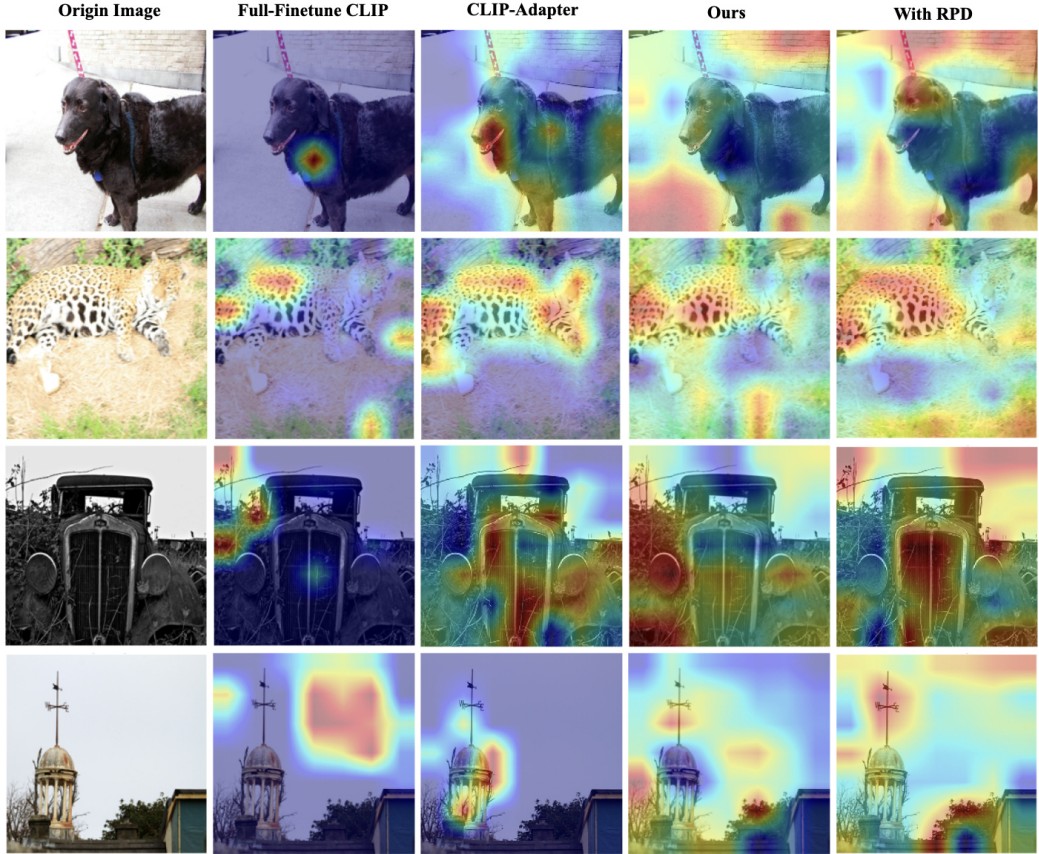

*Figure 11.* Visualization of Gram-Cam Map of models. Our CMPA enables the model's attention to focus more intensively on perceptually relevant regions, while the RPD further refines and strengthens this perceptual emphasis.

### F.2. Perceptual Fidelity and Plug-and-Play Generality of RPD

To demonstrate that our Residual-Enhanced Perceptual Downsampling (RPD) better preserves perceptual quality during downsampling, we evaluate the upsampled results against the original high-resolution images using two full-reference metrics: SSIM and LPIPS (higher SSIM and lower LPIPS indicate better structural and perceptual fidelity).

Results are reported as medians in Table 18 across several widely used IQA datasets.

*Table 18.* Median SSIM and LPIPS after upsampling to original resolution. **Bold** indicates the top results.

| Method | LIVE | | CSIQ | | TID2013 | | KADID-10k | | KonIQ-10k | | LIVEC | | SPAQ | | FLIVE | |
|---|---|---|---|---|---|---|---|---|---|---|---|---|---|---|---|---|
| | SSIM | LPIPS | SSIM | LPIPS | SSIM | LPIPS | SSIM | LPIPS | SSIM | LPIPS | SSIM | LPIPS | SSIM | LPIPS | SSIM | LPIPS |
| Bicubic | 0.928 | 0.257 | 0.852 | 0.223 | 0.861 | 0.125 | 0.893 | 0.099 | 0.854 | 0.262 | 0.913 | 0.210 | 0.811 | 0.545 | 0.880 | 0.138 |
| Lanczos | 0.920 | 0.264 | 0.847 | 0.230 | 0.857 | 0.120 | 0.890 | 0.095 | 0.852 | 0.265 | 0.909 | 0.156 | 0.811 | 0.546 | 0.878 | 0.135 |
| Ours | **0.948** | **0.249** | **0.869** | **0.210** | **0.875** | **0.112** | **0.904** | **0.090** | **0.871** | **0.241** | **0.921** | **0.153** | **0.819** | **0.504** | **0.895** | **0.125** |

Compared to traditional Bicubic and Lanczos interpolation, RPD consistently achieves higher SSIM and lower LPIPS across all datasets. This indicates superior structural preservation and reduced perceptual distance to the original image, confirming the effectiveness of RPD in compensating for perceptual information lost during downsampling.

To further verify that RPD provides more perceptually faithful downsampled images, we evaluate the plug-and-play generality of RPD by applying it as a simple pre-processing step to two off-the-shelf no-reference IQA models (NIQE and MANIQA, from the pyiqa library (Chen & Mo, 2022)) in zero-shot settings. The results are shown in Table 19. Applying RPD as a plug-and-play preprocessing step consistently improves the zero-shot performance of both NIQE and MANIQA across all evaluated datasets. These gains demonstrate the strong generality of RPD: it enhances perceptual alignment for existing NR-IQA models without any retraining or fine-tuning, highlighting its ability to produce downsampled images that

are more perceptually representative of the original content.

*Table 19.* Zero-shot NR-IQA performance (SRCC / PLCC) with and without RPD pre-processing. **Bold** entries indicate the top results. The upper block reports results evaluated by NIQE, while the lower block corresponds to MANIQA.

| Method | LIVE | | CSIQ | | TID2013 | | KADID-10k | | KonIQ-10k | | LIVEC | | SPAQ | | FLIVE | |
|---|---|---|---|---|---|---|---|---|---|---|---|---|---|---|---|---|
| | SRCC | PLCC | SRCC | PLCC | SRCC | PLCC | SRCC | PLCC | SRCC | PLCC | SRCC | PLCC | SRCC | PLCC | SRCC | PLCC |
| **NIQE-Zero-shot Evaluation** | | | | | | | | | | | | | | | | |
| Bicubic | 0.206 | 0.214 | 0.283 | 0.497 | 0.667 | 0.803 | **0.567** | 0.623 | 0.158 | 0.299 | 0.157 | 0.381 | −0.306 | −0.169 | 0.493 | **0.609** |
| Lanczos | 0.206 | 0.188 | 0.293 | 0.519 | 0.638 | 0.790 | 0.536 | **0.625** | 0.174 | 0.307 | 0.161 | 0.386 | −0.303 | −0.167 | **0.514** | 0.607 |
| Ours | **0.397** | **0.398** | **0.435** | **0.650** | **0.662** | **0.791** | 0.527 | 0.529 | **0.183** | **0.331** | **0.263** | **0.437** | 0.004 | **0.067** | 0.507 | 0.605 |
| **MANIQA-Zero-shot Evaluation** | | | | | | | | | | | | | | | | |
| Bicubic | −0.025 | −0.027 | 0.662 | 0.688 | 0.503 | 0.562 | 0.477 | 0.503 | 0.698 | 0.763 | 0.776 | 0.795 | 0.834 | **0.831** | 0.339 | 0.405 |
| Lanczos | −0.025 | −0.028 | 0.659 | 0.684 | 0.504 | 0.560 | 0.475 | 0.501 | 0.698 | 0.762 | 0.776 | 0.794 | **0.834** | 0.830 | 0.337 | 0.405 |
| Ours | **−0.012** | **−0.018** | **0.705** | **0.742** | **0.514** | **0.589** | **0.523** | **0.552** | **0.709** | **0.779** | **0.805** | **0.829** | 0.829 | 0.827 | **0.344** | **0.418** |

### F.3. More visual comparison results

Additional qualitative visualization results, as shown in Figure. 12, including more examples of downsampled images under various scenes and distortion types, are provided in the appendix for further reference.

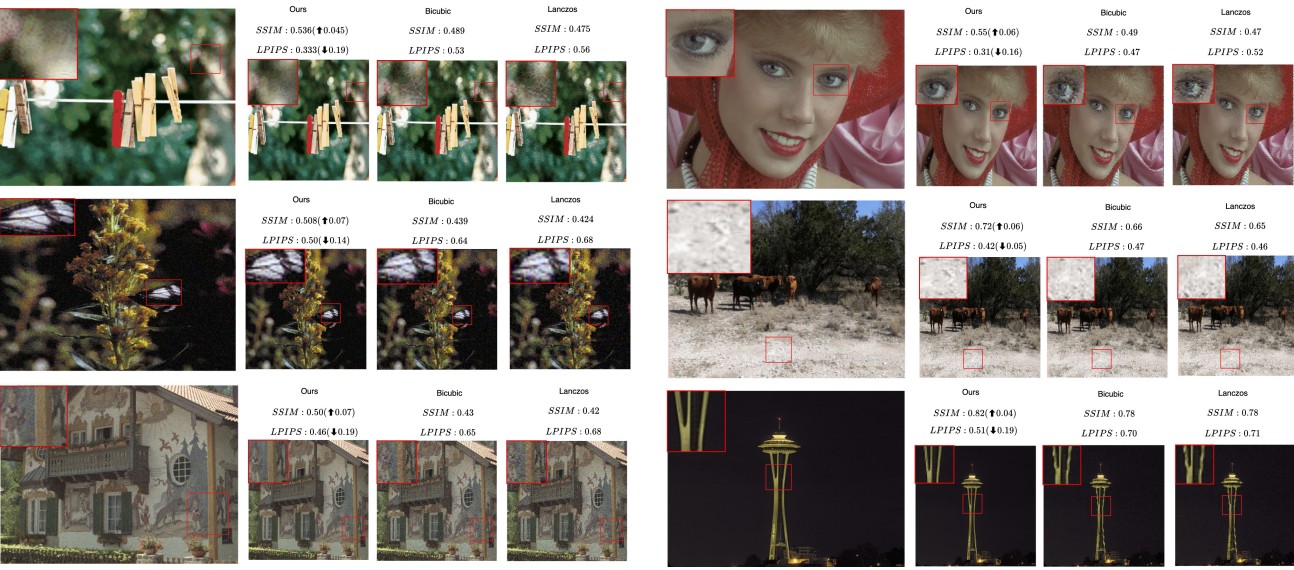

*Figure 12.* More pre-input downsampling results for visual comparison.

## G. Discussion and Limitations

Our experiments and theoretical analysis provide initial evidence that low-dimensional subspaces within feature embeddings exhibit heightened perceptual sensitivity to distortions. This finding aligns qualitatively with prior observations in LoDa (Xu et al., 2024), yet we did not systematically search for the optimal perceptual subspace dimensionality. Instead, we explored a subset of subspace sizes. Determining the theoretically optimal or empirically best dimension remains an important direction for future investigation.

The proposed Residual-Enhanced Perceptual Downscaling (RPD) serves as an intuitive and lightweight compensation strategy for high-frequency information lost during downsampling. However, we acknowledge that RPD is not a universal solution for all resolution rescaling tasks. Specifically, this work primarily compares RPD against standard interpolation baselines and does not extensively evaluate it against modern learnable image resizers or deep-learning-based downsampling methods. This choice was motivated by our goal to maintain a plug-and-play preprocessor without introducing significant

computational overhead or inference latency, which are common drawbacks of deep-learning-based scalers. Additionally, because RPD explicitly injects high-frequency residuals, it may introduce minor perceptual confusion on certain high-frequency-attenuating distortions, such as Pink Gaussian Noise. Consequently, RPD should be viewed as a practical, first-order approximation rather than an optimal perceptually consistent downsampling method. In future work, we plan to investigate more sophisticated, perception-consistent downsampling architectures using lightweight deep learning techniques, aiming to achieve a better balance between input fidelity and computational efficiency in NR-IQA pipelines.

