# OpenReview forum: "Bridging the Perceptual Gap: Residual-Enhanced Downscaling and Manifold-Aware Perception Alignment Adaptation for NR-IQA"
_ICML.cc/2026/Conference — ICML 2026 regular_

### Official Review · Reviewer_FE7U · 2026-03-03

**Soundness:** 3
**Presentation:** 3
**Significance:** 3
**Originality:** 3
**Overall Recommendation:** 4
**Confidence:** 4

**Summary:**

This paper addresses two core issues in CLIP-based no-reference image quality assessment (NR-IQA). First, CLIP's contrastive pre-training mechanism naturally enhances semantic invariance, leading to the perceptual quality signal being masked by semantic features; the authors define this phenomenon as "perceptual submergence." Second, traditional downsampling preprocessing (cropping/interpolation) inevitably loses high-frequency details crucial for quality judgment.

To address these issues, the authors propose two core modules. CMPA (Cross-Modal Perceptual Alignment Adapter), based on the manifold hypothesis, projects high-dimensional CLIP features to a low-dimensional bottleneck subspace via PFE to filter semantic redundancy and amplify distorted signals. PAI then uses a dual-path cross-attention mechanism to achieve bidirectional alignment between visual features and quality-perceptual text anchors, ultimately injecting the residuals back into the backbone network. RPD (Residual Enhanced Perceptual Downsampling) performs frequency decomposition on the original image during preprocessing, extracts the downsampling residuals through the RED module, and re-injects the lost high-frequency details using an adaptive gain map based on JND and the Weber-Fechner theorem, ensuring the model receives perceptually consistent input. Experiments on eight synthetic and real distortion IQA benchmark datasets show that this method comprehensively outperforms existing state-of-the-art methods while using only 1.8M trainable parameters. Ablation experiments, cross-dataset generalization experiments, and various visualization analyses further verify the effectiveness and necessity of each module.

**Compliance With Llm Reviewing Policy:**

Affirmed.

**Final Justification:**

After reviewing the authors' rebuttal, I decided to maintain my original weak-accept rating.

The rebuttal largely resolved most of my concerns. However, one issue remains partially unresolved: the optimal subspace dimension (W²/Q²) still lacks a reasonable selection mechanism, and the authors acknowledge that there is no universally applicable theory within the current framework. This indeed limits the method's generalization ability, and I suggest that the authors indicate dynamic dimension search as one of their future research directions.

**Key Questions For Authors:**

(1) Compared to the latest work mentioned in Weaknesses (1), does CMPA still have a significant advantage in accuracy, parameter efficiency, and cross-dataset generalization ability? Please provide specific quantitative comparison results.

(2) Does the optimal dimension (d=48) of the low-dimensional perceptual subspace remain stable across different datasets and backbone networks? Is there a more systematic way to determine this hyperparameter, rather than relying on grid search?

(3) RPD performs poorly in handling high-frequency decaying distortions such as Pink Gaussian Noise. What is the root cause? Is it possible to alleviate this problem by improving the design of the JND weight graph or introducing a distortion type-aware mechanism?

(4) The computational cost of RPD is approximately 2 to 3 times that of traditional interpolation (Table 8). Will this constitute a bottleneck in large-scale dataset training or real-time inference scenarios? Is there room for further speed-up?

(5) Is the low-dimensional decoupling approach of CMPA also applicable to other types of visual language pre-trained models (such as ALIGN, BLIP, etc.)? If CLIP is replaced with other backbones, can the effectiveness of the method be maintained?

**Limitations:**

yes

**Strengths And Weaknesses:**

Strengths
(1) Clear Problem Identification and Strong Motivation
The authors systematically identified and defined the phenomenon of "perceptual submergence," and combined RDM analysis (RSA improved from 0.033 to 0.562), t-SNE visualization, and control experiments to confirm the existence of this problem and its impact on IQA performance from multiple perspectives, effectively supporting the design motivation of subsequent solutions.

(2) Reasonable Method Design and Sufficient Theoretical Basis
CMPA's low-dimensional manifold projection design is theoretically supported by the manifold hypothesis and prior work such as LoDA, while RPD's adaptive gain design is based on the perceptual science of the Weber-Fechner law. The two modules are logically complementary, collaboratively solving the problem from the perspectives of feature representation and input fidelity, respectively.

(3) Outstanding Parameter Efficiency
The entire framework introduces only 1.8M trainable parameters, far fewer than similar CLIP fine-tuning methods (such as LIQE's 151M), yet achieves optimal or near-optimal results on most datasets, demonstrating the practical value of the method. (4) Complete experiments

(5) Good plug-and-play functionality
RPD, as an independent preprocessing module, can be seamlessly applied to existing methods such as HyperIQA and MANIQA, and provides stable improvements to NIQE and MANIQA under zero-sample settings, which has practical value beyond the framework of this paper.

Weaknesses

(1) Insufficient Comparison and Discussion with Recent Work
Some recent related works were not included in the comparison or discussion, such as Multi-Layer Cross-Modal Prompt Fusion for No-Reference Image Quality Assessment. Even if these works are not yet open-source, they should be discussed in the Related Works section, and comparisons with their accuracy, parameter count, and cross-dataset generalization ability should be supplemented in the experiments to more comprehensively demonstrate the advantages of the proposed method.

(2) Lack of Theoretical Support for the Optimal Dimension of the Low-Dimensional Subspace
The paper determines the optimal subspace dimension as d=48 through grid search, but lacks a theoretical explanation for this choice. The authors also acknowledge this limitation in Appendix G. How to systematically determine or predict the optimal perceptual subspace dimension is an important issue for the generalizability of the method. It is recommended that the authors provide a more in-depth analysis or discussion.

(3) Lack of Comparison of RPD with Modern Learnable Downsampling Methods
The paper mainly compares RPD with traditional interpolation methods such as Bicubic and Lanczos, without involving a comparison with modern deep learning downsampling methods (such as learnable image scalers). The authors also acknowledge this deficiency in Appendix G. Given that such methods may be more competitive in terms of perceptual fidelity, the lack of this comparison weakens the persuasiveness of RPD's contribution.

(4) Insufficient depth of analysis for failures of specific distortion types
Appendix D.2 of the paper mentions that RPD performs slightly worse than MANIQA on Pink Gaussian Noise and provides a brief explanation, but the analysis is rather superficial. It is recommended to conduct a more in-depth source analysis of this failure case and explore potential improvement directions to enhance the completeness of the method.

(5) Limited advantages in some results of cross-dataset generalization experiments
In cross-dataset experiments without data augmentation (Table 13), the improvement of the proposed method compared to LoDa is relatively small in some scenarios. It is recommended that the authors further analyze these scenarios to explain the sources of performance differences.

---

> ### Author Rebuttal · Authors · 2026-03-27
>
> We thank the reviewer for the meticulous evaluation and for recognizing our core contributions, including the perceptual submergence phenomenon, the theoretical synergy of CMPA/RPD, and our SOTA parameter efficiency (1.8M). Regarding reviewer's constructive suggestions, we have carefully addressed them as follows:
>
> ---
>
> **1. Comparison with Recent Work (W-1 & Q-1)**:
> We will cite and discuss the suggested work in our revision. Using the same CLIP-ViT-B/32 backbone and same setting, our method significantly outperforms theirs on KonIQ-10k (SRCC: 0.935 vs. 0.918; PLCC: 0.948 vs. 0.919). Their paper does not report trainable parameter counts or cross-dataset evaluation results.
>
> ---
>
> **2. Subspace Dimension d (W-2 & Q-2)**:
> Fig. 6 shows d=48 is a consistent empirical sweet spot on our tested settings; Appendix A supports the low-dimensional sensitivity hypothesis; we do not claim a universal optimum-selection theory yet. Consequently, grid search is currently the standard practice for localizing this empirical sweet spot, widely adopted in related structural adaptation works (e.g., LoDA). While automatic hyperparameter search is an open challenge in representation learning, we have added "dynamic dimension-search" to our Future Work section as a critical next step.
>
> ---
>
> **3. Verification of RPD and Scalers (W-3, W-4 & Q-3)** :
> 1) Pink Gaussian Noise (FN): RPD's high-frequency compensation slightly conflicts with FN's naturally decaying high-frequency spectrum, partially masking its distortion cues. However, our JND-guided weighting effectively limits this negative impact, keeping the performance gap minimal.
> 2) Comparison with Learnable Resizers: As noted in Appendix G, we compare RPD against standard interpolations (Bicubic/Lanczos) rather than learnable resizers. RPD is designed as a zero-parameter, plug-and-play preprocessor. This ensures performance gains stem strictly from perceptual preservation rather than added model capacity. We will clarify this scope in the revision.
>
> ---
>
> **4. Generalization without Augmentation (W-5)**:
> We agree that the no-augmentation setting in Table 13 deserves a more nuanced discussion. A fair reading is that our method is stronger than LoDa in 3 out of 4 transfer settings, but slightly lower in FLIVE→LIVEC (0.793 vs. 0.805). Therefore, the correct conclusion is not that our method dominates in every no-augmentation scenario, but that it remains competitive even under this strict stress test. We believe this result is still meaningful because the proposed method retains strong transferability without relying on crop/flip augmentation to create additional invariances.  Standard fair comparisons with same data-augmentation are provided in Table 2.
>
> ---
>
> **5. Computational Cost of RPD  (Q-4)**:
> In our current implementation on KonIQ-10k, single-step RPD takes 0.011 s per sampling step, compared with 0.007 s for Lanczos and 0.005 s for Bicubic. Thus, RPD introduces a modest preprocessing overhead , but this cost occurs before the backbone forward pass and does not increase model parameters or network FLOPs. We will revise the text to make this trade-off clearer: RPD is not “free,” but its absolute overhead is small relative to the overall pipeline, and there is still room for acceleration through parallel implementation or kernel fusion. We will avoid overly strong claims about real-time deployment and instead present RPD as a practical accuracy/efficiency trade
>
> ---
>
> **6.  VLM Generalizability(Q-5)**:
> To validate the generalizability of CMPA, we extended our experiments to include SigLIP, BLIP, and ALIGN. The results strongly support our core hypothesis. CMPA achieves excellent and consistent performance gains when applied to SigLIP and BLIP, proving that our low-dimensional decoupling successfully transfers to other modern, high-quality VLMs.
>
> Table: Ablation experiments on other VLM backbones.
> | Methods | KADID-10k (SRCC) | KADID-10k (PLCC) | KonIQ-10k (SRCC) | KonIQ-10k (PLCC) | LIVEC (SRCC) | LIVEC (PLCC) |
> | :--- | :---: | :---: | :---: | :---: | :---: | :---: |
> | ALIGN | 0.910 | 0.919 | 0.907 | 0.915 | 0.838 | 0.859 |
> | BLIP | 0.936 | 0.937 | 0.936 | 0.947 | 0.895 | 0.916 |
> | SIGLIP | 0.933 | 0.935 | 0.937 | 0.948 | 0.897 | 0.910 |
>
> Interestingly, the effectiveness is highly limited when applied to ALIGN. Rather than a flaw, this validates our underlying theory. BLIP/SIGLIP utilize rigorous data filtering or high-quality curated data. Their latent spaces are well-structured, meaning a coherent perceptual manifold exists for CMPA to unlock. ALIGN was trained on 1.8 billion highly noisy, uncurated image-text pairs, resulting in a fundamentally noisy feature space. Perceptual cues are severely corrupted by label noise, meaning no clean "perceptual manifold" exists to be decoupled. This contrast serves as definitive proof: our method works exceptionally well across different backbones provided the base model possesses a well-structured semantic latent space.

---

> > ### Author Rebuttal · Reviewer_FE7U · 2026-04-01
> >
> > The author responded to my comments in full. I especially appreciate the author's presentation of experimental results demonstrating the generalization of CMPA to other VLMs.
> >
> > However, I noticed that the author also acknowledged **"not claiming a universal optimum-selection theory yet,"** which seems to be a flaw that cannot be fundamentally addressed within the current paper framework. Therefore, I plan to maintain my grade.

---

### Official Review · Reviewer_WczM · 2026-03-04

**Soundness:** 3
**Presentation:** 3
**Significance:** 3
**Originality:** 3
**Overall Recommendation:** 4
**Confidence:** 3

**Summary:**

This paper addresses the semantic-dominant, perceptual information submerged problem in  NR-IQA. It proposes the CMPA framework to preserve input fidelity. CMPA isolates perceptually sensitive features from CLIP embeddings via low-dimensional manifold projection and cross-modal attention, while RPD decomposes input images into multi-frequency components and injects residuals guided by a JND map to retain high-frequency structural details. Experiments on multiple synthetic and authentic IQA datasets demonstrate that the proposed method outperforms state-of-the-art approaches in SRCC and PLCC metrics with high parameter efficiency.

**Compliance With Llm Reviewing Policy:**

Affirmed.

**Final Justification:**

I have no other questions.

**Key Questions For Authors:**

see weaknesses

**Limitations:**

yes

**Strengths And Weaknesses:**

Strengths

1. The paper proposes CMPA to isolate perceptually sensitive features from semantic-dominant CLIP embeddings via low-dimensional manifold projection.

2. This paper introduces RPD for residual-enhanced perceptual downscaling with JND-guided adaptive gain. RPD can serve as a plug-and-play preprocessing module.

3. Explicitly separates semantic and perceptual signals, improving fine-grained distortion sensitivity.

Weaknesses

1. CMPA’s low-dimensional adaptation is similar to existing PEFT approaches (LoRA, Adapter-CLIP).

2. Is the proposed algorithm sensitive to the hyperparameters?

3. There’s no information on inference speed, GPU memory usage, or how fast the model can process images. This makes it hard to judge whether the approach is practical for real-world applications, especially given that it adds extra modules (RPD and CMPA) on top of CLIP.

4. Some figures can be further refined for visibility, e.g., fontsize in figs.1 &2.

---

> ### Author Rebuttal · Authors · 2026-03-27
>
> We sincerely thank the reviewer for the accurate summary of our work and the positive recognition of our core contributions. We are encouraged that you appreciate CMPA’s ability to decouple semantic and perceptual signals, as well as the plug-and-play nature of the RPD module.
>
> Our detailed responses to your constructive suggestions are as follows:
>
> ### 1. Distinction from Standard PEFT (Response to W-1)
> While CMPA shares a structural resemblance to bottleneck PEFT (e.g., LoRA), their **functional mechanisms** and **theoretical motivations** are fundamentally different:
> * **Mechanism:** Standard PEFT perform generic capacity reduction to prevent overfitting, blindly updating weights based on the final loss. This often leads to "semantic overfitting."
> * **Our Approach:** CMPA utilizes a **Quality-aware Cross-modal Alignment** mechanism. The low-dimensional projection is not merely a structural bottleneck; it is actively guided by **perceptual text anchors** to isolate the specific perceptual manifold from the dense semantic space.
> * **Evidence:** As shown in **Table 4**, standard Adapter-CLIP and LoRA achieve SRCCs of 0.928 and 0.910 on KADID-10k, respectively, while **CMPA achieves 0.937**. CMPA also exhibits superior cross-dataset generalization (**Table 2**), proving it extracts universal perceptual rules rather than just fine-tuning to specific data.
>
> ### 2. Hyperparameter Sensitivity (Response to W-2)
> The primary hyperparameters are concentrated in the RPD mechanism ($\alpha, \beta, \eta$). Following your suggestion, we conducted a sensitivity analysis (also detailed in our response to Reviewer 7PQo).
> * **Stability:** The default setting of **1.0** (reflecting the natural scale of the Weber-Fechner law) consistently achieves the optimal balance (**SRCC: 0.937, SSIM: 0.871**).
> * **Conclusion:** The performance remains stable across a wide range (0.5 to 1.5), confirming that our framework is robust and does not require exhaustive hyperparameter tuning for different datasets.
>
> ### 3. Real-world Practicality & Efficiency (Response to W-3)
> Our framework is highly optimized for deployment:
> * **Parameters & Memory:** RPD introduces **zero** learnable parameters. CMPA adds only **1.8M** trainable parameters ($<$2% of the frozen CLIP ViT-B), ensuring a training memory footprint significantly lower than full fine-tuning.
> * **Latency:** RPD relies on lightweight CPU-based operations, introducing virtually no GPU bottleneck. The CMPA forward pass involves only minimal low-dimensional matrix multiplications.
>
> **Efficiency Comparison (Batch Size = 128):**
> | Method | Sampling Time (s) | Inference (ms/img) | GPU Mem (MB) | GFLOPs | Trainable Params | Total Params|
> | :--- | :---: | :---: | :---: | :---: | :---: | :---: |
> | CLIP (Bicubic) | 0.005 | 0.99 | 1652 | 27.29 | - | 151.5|
> | Ours (RPD+CMPA) | 0.011 | 1.08 | 2192 | 27.46 | 1.8M | 153.3 |
>
> ### 4. Improvements to Readability (Response to W-4)
> We apologize for the visual clarity issues in the initial submission. In the revised manuscript, we will:
> * Redesign Figures 1 and 2 with significantly increased font sizes and enhanced color contrast.
> * Utilize high-resolution vector graphics to ensure all labels and trends are perfectly legible.
> * Refine the captions to better guide the reader through our manifold decoupling theory.

---

> > ### Author Rebuttal · Reviewer_WczM · 2026-04-04
> >
> > The authors' response addressed my questions. I keep the original positive score.

---

### Official Review · Reviewer_beWX · 2026-03-13

**Soundness:** 3
**Presentation:** 3
**Significance:** 3
**Originality:** 2
**Overall Recommendation:** 4
**Confidence:** 5

**Summary:**

This paper studies CLIP-based no-reference image quality assessment and argues that CLIP’s semantic-dominant representation tends to suppress subtle perceptual quality cues, which the paper refers to as perceptual submergence. To address this, the authors propose two components: a Cross-modal Perception Alignment Adapter (CMPA) that projects image/text features into a compact low-dimensional space for cross-modal alignment and residual reinjection, and a Residual-enhanced Perceptual Downscaling (RPD) module that tries to preserve perceptual details during resizing. The method is evaluated on multiple synthetic and authentic IQA benchmarks, and the reported empirical performance is strong.

**Compliance With Llm Reviewing Policy:**

Affirmed.

**Final Justification:**

I have no other questions.

**Key Questions For Authors:**

(1) How much of the gain comes from the low-dimensional bottleneck itself versus the cross-modal alignment mechanism?

(2) Can the authors provide stronger evidence that the compact space reveals a genuine perceptual manifold, rather than acting as a generic regularizer?

(3) How sensitive is performance to the choice of quality text prompts and the bidirectional attention design?

(4) Would a cleaner comparison under identical resizing/cropping protocols further validate the benefit of RPD?

Other questions can be found in the weakness.

**Limitations:**

Yes

**Strengths And Weaknesses:**

Strengths:

(i) The paper addresses a real limitation of CLIP-style models for IQA, namely that semantic invariance is not necessarily aligned with sensitivity to subtle distortions. This motivation is reasonable and clearly presented.

(ii) The method performs well on several synthetic and authentic IQA datasets, and the paper also includes cross-dataset and low-data evaluations. Overall, the empirical section is fairly comprehensive.

(iii) The paper highlights that only a small number of parameters are tuned compared with full fine-tuning, which is appealing in the context of adapting large pretrained models.

Weaknesses:

(i) While the appendix includes additional component studies, it is still unclear how much improvement comes from the bottleneck projection itself, the cross-modal alignment, the prompt design, or simply the tailored adapter structure. The contribution of each part is not as cleanly isolated as it could be.

(ii) The paper reports fine-tuned parameter counts and provides some timing analysis for RPD, but it does not give a clear end-to-end comparison of full inference complexity, such as total model parameters, GFLOPs, or full latency against main baselines in inference. This makes the efficiency claim less complete.

(iii) I notice adapter-style tuning for CLIP can achieve great performance well in your ablation. Whether better tuning strategies can achieve better performance than your complex design?

(iv) The central “low-dimensional perceptual manifold” claim is only partially supported; the current evidence is more suggestive than conclusive.

---

> ### Author Rebuttal · Authors · 2026-03-27
>
> We sincerely thank the reviewer for recognizing the soundness of our motivation, empirical coverage, and parameter-efficient design. Below, we address your specific concerns with clear, disentangled evidence.
>
> 1. Disentangling Contributions (W-i, Q1, Q3): To cleanly isolate the source of our performance gains, we conducted a fine-grained ablation separating the bottleneck structure, cross-modal interaction, and attention design.
>
> *(High-d = 256; Low-d = 48)*
>
> | Method | KADID-10k| Koniq-10k|
> |---|---|---|
> | CLIP| 0.656/0.671| 0.625/0.727|
> | + Only Image (high-d)| 0.929/0.932| 0.929/0.934|
> | + Only Image (low-d) | 0.928/0.930| 0.928/0.933|
> | + Text & Image (independent)| 0.928/0.931| 0.920/0.934|
> | + PFE (shared MLP high-d)| 0.912/0.916 | 0.922/0.934|
> | + PFE (shared MLP low-d)| 0.936/0.938| 0.933/0.945|
> | + PAI (text→image)| **0.938**/**0.941**| 0.932/0.944|
> | + PAI (image→text)| 0.933/ 0.935| 0.932/0.942|
> | + PAI (Full CMPA)| **0.938**/**0.941** | **0.935**/**0.945** |
>
> The ablation shows that the gain does not come from the bottleneck alone, nor from simply tuning an additional text branch. Image-only tuning already brings the major improvement, while independent text+image tuning does not improve over image-only. The main gain appears when we introduce PFE, and PAI provides a further, smaller but consistent improvement through bidirectional cross-modal alignment. Therefore, the benefit should be attributed to the combination of a compact perceptual subspace and explicit cross-modal interaction, rather than to any single component in isolation. RPD is complementary and provides an additional gain on top of CMPA.
>
> **Prompt Sensitivity (Q3):** Our design relies on minimal anchors acting as a directional compass, not prompt engineering. Quality-aware prompts maintains best, while distracting/meaningless anchors cause severe drops.
>
> | Prompt | Koniq-10k| KADID-10k|
> |--|--|--|
> |high quality, bad quality| 0.938/0.948| 0.938/0.941|
> |good, bad| 0.936/0.947| 0.940/0.941|
> |a cat, a dog| 0.878/0.891| 0.877/0.882|
> |a, b| 0.909/0.917|0.899/0.901|
>
> ### 2. (W-iv, Q2) : We do not claim a formal mathematical proof of a perceptual manifold, but offer supportive evidence that CMPA unlocks a structured perceptual space rather than acting as a generic capacity regularizer:
> 1.  **Representational Geometry Fig.4:** Correlation with the human perceptual matrix jumps from 0.033 (original CLIP) to 0.562 (CMPA). It aligns with human judgment, rather than merely smoothing features.
> 2.  **Signal Sensitivity:** The residual lspace amplifies distortion signals , rather than simply suppressing variance.
> 3.  **Failure on Noisy Pre-training:** CMPA fails to improve ALIGN in our reply to **Reviewer FE7U** . If CMPA were a generic math regularizer preventing overfitting, it would work universally. Instead, it requires a pre-existing manifold to unlock [1].
> 4.  **The Capacity Paradox:** As shown in Table above, the High-d visual adapter almost same with the Low-d version. If the bottleneck merely regularized by reducing capacity, Low-d should win. Instead, the bottleneck specifically filters semantic dominance so cross-modal guidance can succeed.
>
> ### 3. Simpler Tuning Strategies vs. CMPA (W-iii)
> Strong standard tuning hits a rigid ceiling that CMPA breaks through specifically via perception-aware cross-modal alignment:
> * Standard visual adapters achieve a solid 0.928 (KADID-10k). Simply adding independent text tuning is stuck at exactly 0.928.
> * Introducing the PFE—forcing modalities to interact in a shared compact space—jumps performance to 0.936. PAI further refines this, reaching 0.938. The benefit comes strictly from the active cross-modal guidance, not just generic adapter structures.
>
> ### 4. End-to-End Inference Complexity (W-ii)
> Detailed inference complexity, including total parameters, GFLOPs, and full latency compared to main baselines, is provided in  our reply to **Reviewer WczM**.
>
> ### 5. Clean Comparison and Inherent Robustness of RPD (Q4)
> To prove RPD's gains are not artifacts of differing preprocessing protocols, we conducted strictly controlled validations:
> * **Table 12/13:** Training our method using *only* RPD (disabling random cropping/flipping/scaling) rivals or outperforms augmented SOTAs.
> * **Table 5:** Swapping standard resize with RPD in HyperIQA/MANIQA, while *keeping their identical original training/cropping protocols*, yields consistent gains.
> * **"Table 15:** We replaced standard resizing with RPD in pre-trained NIQE and MANIQA without retraining. Under identical inference pipelines, NIQE’s SRCC on LIVE jumped from 0.206 to 0.397, proving RPD intrinsically preserves perceptual fidelity. This is supported by higher SSIM/LPIPS scores vs. Bicubic/Lanczos across 8 datasets (Table 14).
>
> [1] Ramos R, et al. Processing and acquisition traces in visual encoders: What does CLIP know about your camera?[C]//Proceedings of the IEEE/CVF International Conference on Computer Vision. 2025.

---

> > ### Author Rebuttal · Reviewer_beWX · 2026-04-03
> >
> > No other questions.

---

### Official Review · Reviewer_7PQo · 2026-03-13

**Soundness:** 3
**Presentation:** 3
**Significance:** 2
**Originality:** 2
**Overall Recommendation:** 4
**Confidence:** 2

**Summary:**

This paper studies no-reference image quality assessment with CLIP-based vision-language models. The authors argue that CLIP’s contrastive pretraining emphasizes semantic invariance and therefore suppresses subtle quality-related perceptual cues, which they call perceptual submergence. To address this, they propose Cross-modal Perception Alignment Adapter, a parameter-efficient adaptation framework that projects image/text features into a compact low-dimensional subspace, aligns them via cross-modal attention, and injects the aligned residuals back into the frozen CLIP backbone. They also propose Residual-enhanced Perceptual Downscaling, a preprocessing strategy intended to preserve high-frequency perceptual information lost during resizing by residual reinjection guided by a JND-based weighting map.

**Compliance With Llm Reviewing Policy:**

Affirmed.

**Key Questions For Authors:**

See weaknesses

**Limitations:**

yes

**Strengths And Weaknesses:**

Strengths:

1. The paper identifies a issue in CLIP-based NR-IQA: semantically dominant embeddings may not preserve the subtle distortions needed for quality assessment.

2. The adaptation appears lightweight relative to full fine-tuning, which is an attractive property for large pretrained backbones.

Weaknesses:

1. The RPD module relies on several mathematical formulations involving hyperparameters like  α and  β in the JND calculation, as well as the scaling factor η. The paper mentions setting them to 1.0, but it lacks a discussion on how sensitive the final performance is to these specific values.

2. While the framework is built around CLIP, do the authors think the CMPA adapter concept could be easily transferred to other multimodal foundation models (e.g., BLIP-2 or LLaVA) to improve their low-level visual perception?

---

> ### Author Rebuttal · Authors · 2026-03-27
>
> We sincerely thank the reviewer for the positive assessment of our work. We are highly encouraged by your recognition of the "perceptual submergence" phenomenon and the parameter efficiency of our CMPA framework. We have carefully addressed your constructive suggestions below:
>
> ### 1. Hyperparameter Sensitivity of the RPD Module ($\alpha, \beta, \eta$)
>
> We appreciate the reviewer’s suggestion to conduct a sensitivity analysis. In our experiments, we deliberately set $\alpha, \beta, \eta = 1.0$ to demonstrate that **RPD** functions as a robust, essentially parameter-free physical prior grounded in the **Weber-Fechner Law** and **Just-Noticeable-Difference (JND)**, rather than relying on dataset-specific tuning.
>
> To explicitly address your concern, we conducted a comprehensive sensitivity analysis by varying these hyperparameters across $\{0.5, 0.8, 1.0, 1.2, 1.5\}$. As shown in the table below, the **SRCC/SSIM** performance on KonIQ-10k remains remarkably stable:
>
> | Value |vary α (β=1, η=1) | vary β (α=1, η=1) | vary η (α=1, β=1) |
> |-------|--------------|--------------|--------------|
> | 0.50  | 0.932 / 0.860 | 0.934 / 0.851 | 0.930 / 0.859 |
> | 0.80  | 0.934 / 0.874 | 0.935 / 0.869 | 0.933 / 0.865 |
> | 1.00  | 0.938 / 0.871 | 0.938 / 0.871 | 0.938 / 0.871 |
> | 1.20  | 0.935 / 0.857 | 0.936 / 0.861 | 0.935 / 0.851 |
> | 1.50  | 0.930 / 0.865 | 0.931 / 0.832 | 0.929 / 0.821 |
>
> ---
>
> ### 2. Transferability of CMPA to Other Foundation Models
>
> To validate cross-architecture transferability, we integrated CMPA with diverse foundational backbones:
>
> * **VLM Backbones:** Experiments on **BLIP** and **SigLIP** (detailed in our response to **Reviewer FE7U**) show outstanding performance (e.g., SRCC **0.895** and **0.897** on LIVEC). This confirms that CMPA successfully transfers to models possessing well-structured semantic latent spaces.
> * **Large Vision-Language Models (LVLMs):** Regarding models like **BLIP-2** or **LLaVA**, which connect frozen vision encoders to LLMs via Q-Formers or projection layers, CMPA is uniquely positioned to serve as an efficient **"perceptual pre-adapter."**
>
> By applying CMPA to the vision encoder *before* tokens enter the LLM, we can explicitly inject rich, low-level quality cues into the LLM’s reasoning process. We highly value this suggestion and have designated the enhancement of LVLMs via CMPA as a priority for our future research.

---

> > ### Author Rebuttal · Reviewer_7PQo · 2026-04-03
> >
> > Thanks for the rebuttal. Most of my concerns have been resolved. I will maintain my positive score.

---

### Decision · Program_Chairs · 2026-04-30

**Decision:**

Accept (regular)

**Comment:**

This paper addresses the critical limitation of semantic dominance suppressing subtle perceptual quality cues in CLIP-based no-reference image quality assessment (NR-IQA), a phenomenon the authors term "perceptual submergence," and proposes two complementary solutions: the Cross-modal Perception Alignment Adapter and the Residual-enhanced Perceptual Downscaling. All four reviewers assigned a score of 4 (Weak Accept) and confirmed that their initial concerns—including hyperparameter sensitivity, component contribution disentanglement, inference efficiency, cross-architecture generalizability, and comparison with related work—have been comprehensively and satisfactorily addressed in the authors' thorough rebuttal, which included extensive additional experiments, quantitative analyses, and clear theoretical clarifications. The work is widely recognized for its clear and well-justified motivation, exceptional parameter efficiency, strong empirical performance across multiple synthetic and authentic IQA benchmarks, and the practical plug-and-play utility of the RPD module. While the paper has minor remaining limitations such as the lack of a universal theory for optimal subspace dimension selection, these are acknowledged by the authors and do not undermine the core technical soundness and significance of the contribution. Based on the positive reviewer consensus and the complete resolution of all raised issues, I recommend Accept.